# Ocean Phosphorus Inventory: Large Uncertainties in Future Projections on Millennial Timescales and its Consequences for Ocean Deoxygenation

Tronje P. Kemena[1], Angela Landolfi[1], Andreas Oschlies[1], Klaus Wallmann[1], Andrew W. Dale[1]

[1]GEOMAR Helmholtz Centre for Ocean Research Kiel, Düsternbrooker Weg 20, 24105, Kiel

*Correspondence to*: Tronje P. Kemena (tkemena@geomar.de)

**Abstract**

Previous studies have suggested that enhanced weathering and benthic phosphorus (P) fluxes, triggered by climate warming, can increase the oceanic P inventory on millennial time scales, promoting ocean productivity and deoxygenation. In this study, we assessed the major uncertainties in projected P inventories and their imprint on ocean deoxygenation using an Earth system model of intermediate complexity for the same business-as-usual carbon dioxide ($CO_2$) emission scenario until the year 2300 and subsequent linear decline to zero emissions until the year 3000. Our set of model experiments under the same climate scenarios but differing in their biogeochemical P parameterizations, suggest a large spread in the simulated oceanic P inventory due to uncertainties in (1) assumptions for weathering parameters, (2) the representation of bathymetry on slopes and shelves in the model bathymetry, (3) the parametrization of benthic P fluxes and (4) the representation of sediment P inventories. Considering the weathering parameters closest to present day, a limited P reservoir and prescribed anthropogenic P fluxes, we find a +30% increase in the total global ocean P inventory by the year 5000 relative to pre-industrial levels, caused by global warming. Weathering, benthic and anthropogenic fluxes of P contributed +25%, +3% and +2% respectively. The total range of oceanic P inventory changes across all model simulations varied between +2% and +60%. Suboxic volumes were up to 5 times larger than in a model simulation with a constant oceanic P inventory. Considerably large amounts of the additional P left the ocean surface unused by phytoplankton via physical transport processes as preformed P. In the model, nitrogen fixation was not able to adjust the oceanic nitrogen inventory to the increasing P levels or to compensate for the nitrogen loss due to increased denitrification. This is because low temperatures and iron limitation inhibited the uptake of the extra P and growth by nitrogen fixers in polar and lower latitude regions. We suggest that uncertainties in P weathering, nitrogen fixation and benthic P feedbacks need to be reduced to achieve more reliable projections of oceanic deoxygenation on millennial timescales.

## 1 Introduction

The oxygen balance in the ocean is regulated by physical supply and the biological consumption. Warming has been found to be a major driver of oceanic oxygen variability, acting via changes in ocean solubility and indirect changes in circulation and biological production and respiration (Battaglia

and Joss, 2018; Levin, 2019; Oschlies et al., 2018; Yamato et al.; 2015). Phosphorus is considered the ultimate limiting nutrient for ocean productivity at global scales (Tyrrell, 1999). Thus, changes in oceanic phosphorus (P) inventories are also hypothesized to substantially affect oceanic oxygen inventories on millennial timescales (Tsandev and Slomp, 2009; Palastanga et al., 2011; Monteiro et al., 2012). Elevated supply of P to the ocean stimulates production and export of organic matter and deoxygenation, which possibly drives more intense oxygen depletion in the oxygen deficient zones and along the continental margins, with release of additional P from sediments turning anoxic (Van Cappellen and Ingall, 1994; Palastanga et al., 2011). Such a positive feedback was discussed for a global warming scenario under present-day conditions (Niemeyer et al., 2017) as well as for large-scale deoxygenation events in the Cretaceous era, the so-called oceanic anoxic events (OAEs) (Tsandev and Slomp, 2009; Monteiro et al., 2012; Ruvalcaba Baroni et al., 2014). For the Cretaceous, it has been suggested that atmospheric carbon dioxide ($CO_2$) concentrations as high as 1000 to 3000 ppmv, driven by enhanced $CO_2$ outgassing from volcanic activity (Jones and Jenkyns, 2001; Kidder and Worsley, 2012), have triggered OAEs (Damsté et al., 2008; Méhay et al., 2009; Bauer et al., 2016). The warmer climate during past OAEs increased weathering on land (Blättler et al., 2011; Pogge von Strandmann et al., 2013), leading to an enhanced supply of nutrients, in particular P, increasing the oceanic nutrient inventory and driving the positive feedback mentioned above. Furthermore, the enhanced release of P from sediments were suggested to maintain high levels of productivity in the Cretaceous ocean (Mort et al. 2007; Kraal et al. 2010), which would contribute to the development of OAEs. Evidence in the palaeo record indicates that the Earth has experienced several OAEs with large-scale anoxia, euxinia and mass extinctions (Kidder and Worsley, 2010).

Could such OAEs also appear in the near future under contemporary global warming? High $CO_2$ concentrations in the atmosphere seem to be one driver for initiating OAEs and ocean deoxygenation. Projected anthropogenic $CO_2$ emissions may lead to atmospheric $CO_2$ concentrations exceeding 1000 ppmv at the beginning of the $22^{nd}$ century if emissions continue to increase in a business-as-usual scenario (Meinshausen et al., 2011). Although anthropogenic $CO_2$ emissions occur over a short period compared to the long-term and relatively constant volcanic $CO_2$ emissions during OAEs (Kidder and Worsley, 2012), elevated atmospheric $CO_2$ concentrations will persist for many millennia (Clark et al., 2016). This may provide the conditions for long-term climate change and large-scale deoxygenation. There is thus some concern that anthropogenic $CO_2$ emissions could potentially trigger another OAE (Watson et al., 2017). Yet, Kidder and Worsley (2012) argue that emissions of global fossil fuel reserves are insufficient to drive a modern OAE, but may instead lead to widespread suboxia.

During climate warming, ocean productivity could switch from P to nitrogen (N) limitation (Saltzman, 2005). N limitation could arise from enhanced denitrification in a more anoxic ocean, but at the same time low N to P ratios would be expected to stimulate $N_2$-fixation by diazotrophs (Kuypers et al., 2004). $N_2$-fixation in regional proximity with OMZs can lead to net N losses due to mass balance constraints (Landolfi et al., 2013), which may even reverse the net effect of $N_2$-fixation on the nitrogen inventory. Recently, Niemeyer et al. (2017) showed in a model study that P weathering and sedimentary P release in a business-as-usual $CO_2$-emission (RCP8.5) scenario could strongly enlarge the marine P inventory and lead to a 4 to 5-fold increase in the suboxic water volume (dissolved

oxygen ($O_2$) concentrations less than 5 mmol m$^{-3}$) on millennial timescales. Here, we build on this study and test the sensitivity of the marine P and $O_2$ inventories to changes in P weathering, benthic and anthropogenic fluxes under the same future scenario on millennial timescales. We aim to provide better constraints on future ocean deoxygenation and assess the biogeochemical feedbacks triggered by

P addition. In Sect. 2 we present the experimental design and the model parameterizations of continental P weathering and of benthic P release. In Sect. 3 we assess uncertainties in P fluxes due to different assumptions about the P weathering fluxes, different model formulations of benthic P burial and improved representation of bathymetry and anthropogenic P fluxes. Consequences for deoxygenation and for the biogeochemical cycling of nutrients are discussed.

**2 Model and Experimental Design**

**2.1 Model**

We applied the University of Victoria (UVic) Earth System Model (ESM) version 2.9 (Weaver et al., 2001), which has been used in several studies to investigate ocean oxygen dynamics (Schmittner et al., 2007; Oschlies et al., 2008; Getzlaff et al., 2016; Keller et al., 2016; Landolfi et al., 2017). The UVic

model consists of a terrestrial model based on TRIFFID and MOSES (Meissner et al., 2003), an atmospheric energy-moisture-balance model (Fanning and Weaver, 1996), a sea-ice model (Bitz and Lipscomb, 1999) and the general ocean circulation model MOM2 (Pacanowski, 1996). Horizontal resolution of all model components is 1.8° latitude x 3.6° longitude. The ocean model has 19 layers with layer thicknesses ranging from 50 m for the surface layer to 500 m in the deep ocean. The marine

ecosystem was represented by a NPZD model (Keller et al., 2012). Organic matter transformations (production, grazing, degradation) were parameterized using fixed stoichiometric molar ratios (C:N:P, 106:16:1) and directly related to the production and, in oxygenated waters, utilization of $O_2$ (O:P, 160). When $O_2$ is depleted in the model, organic matter is respired using nitrate ($NO_3^-$) (i.e. microbial denitrification). An $O_2$ concentration of 5 mmol m$^{-3}$ was used as the switching point from aerobic

respiration to denitrification. Sedimentary denitrification was not considered in this model configuration so that water column denitrification and $N_2$-fixation dictate the oceanic N balance. No explicit iron cycle was simulated and iron limitation was approximated with prescribed seasonally varying dissolved iron concentrations (Keller et al., 2012). Parameterizations of benthic and weathering fluxes of P were extended from the study of Niemeyer et al. (2017). A calcium carbonate sediment

model (Archer, 1996) and a parameterization for silicate and carbonate weathering (Meissner et al., 2012) were applied in all simulations. When P weathering and anthropogenic P fluxes were applied (see Sect. 2.2), the global P flux was distributed over all river basins, in every grid box, weighted by river discharge rates.

**2.2 Experimental Design**

Twelve different model simulations were performed to explore the range of uncertainties for the long-term development of the oceanic P inventory (Table 1). Each simulation started from an Earth system state close to equilibrium under preindustrial atmospheric $CO_2$ concentrations, prescribed wind fields

and present-day orbital forcing. Spin-up runs lasting 20,000 simulation years or longer were made for each simulation to reach equilibrium. In the spin-up runs for simulations with benthic P burial (purple and red in Table 1), the marine P inventory was kept constant by instantaneously compensating oceanic P loss (burial) by P weathering fluxes to the ocean. For model simulations without benthic P burial (black and blue in Table 1), one common spin-up run was performed without P weathering fluxes.

All transient simulations started in the year 1765 and ended in the year 5000. Simulations were forced with anthropogenic $CO_2$ emissions (fossil fuel and land use change) according to the extended RCP 8.5 scenario until the year 2300 (Meinshausen et al., 2011), followed by a linear decline to zero $CO_2$ emissions by the year 3000. Warming from non-$CO_2$ greenhouse gases and the effect of sulphate aerosols were prescribed as radiative forcing (Eby et al., 2013). Non $CO_2$-emission effects from land-use change were not considered. The reference simulation (*Ref*) was performed without weathering and without burial fluxes of P, meaning that the P inventory of the ocean remained unchanged. The remaining transient simulations applied either variable climate-sensitive weathering anomalies (without burial) or time-variable burial fluxes (with constant weathering) to the ocean (Table 1).

**2.3 Burial experiments**

The water column model is not coupled to a prognostic and vertically resolved sediment model. Instead, sinking organic matter interacts with the sediment via "transfer functions" (Wallmann, 2010) on a detailed subgrid bathymetry (Somes et al., 2013). Sinking organic matter is partially intercepted at the bottom of each grid box by a sediment layer and the intercepted amount depends linearly on the fractional coverage of the grid box by seafloor. The intercepted organic P is remineralized in accordance with Eq. (1) and Eq. (2), whereby organic C and N are completely remineralized under oxygen or nitrate utilization without any burial.

Fractional coverage of every ocean grid box by seafloor was calculated on each model depth level according to the subgrid bathymetry (Somes et al., 2013). The subgrid bathymetry was inferred from ETOPO2v2[1] (National Geophysical Data Center, 2006). ETOPO2v2 has a horizontal resolution of 2-minutes fine enough to adequately represent continental shelves and slopes. The coarse standard model bathymetry in the UVic model has a horizontal resolution of 1.8° latitude x 3.6° longitude.

P burial in the sediment ($BUR_P$) was determined in every grid box with sediment from the difference between the simulated detritus P rain rate to the sediment ($RR_P$) and the benthic release of dissolved inorganic P from the sediment ($BEN_P$):

$$BUR_P = RR_P - BEN_P \qquad (1)$$

where $RR_P$ is the detritus flux from the ocean (in P units). $BEN_P$ was calculated locally by a "transfer function", which parameterizes sediment/water exchange of P as a function of the rain rate of organic matter and the bottom water $O_2$ concentration. Preferential P release, relative to carbon (C), is observed in sediments overlain by $O_2$-depleted bottom waters (Ingall and Jahnke, 1994). Benthic P release was dependent on the dissolved inorganic carbon release ($BEN_C$) from organic matter degradation in the sediment and the C:P regeneration ratio $r_{C:P}$ (Wallmann, 2010; equation 2):

---

[1] https://www.ngdc.noaa.gov/mgg/global/etopo2.html

$$BEN_P = \frac{BEN_C}{r_{C:P}}. \tag{2}$$

$BEN_C$ was computed (Eq. 3a) as the difference of the carbon rain rate to the sediment ($RR_C$) and a 'virtual' organic carbon burial flux ($BUR_C$). This flux is "virtual" as we do not account for changes in the C inventory and there is no explicit burial of organic C, which is remineralized in the deepest ocean layer. $BUR_C$ is dependent on the simulated organic C rain rate and bathymetry (Flögel et al., 2011). Burial of organic C is more efficient on the shelf and continental margins (Eq. 3b) than for the deep sea (Eq. 3c, sediment below 1000m water depth):

$$BEN_C = RR_C - BUR_C. \tag{3a}$$

$$BUR_C = 0.14 \cdot RR_C^{1.11}, \tag{3b}$$

$$BUR_C = 0.014 \cdot RR_C^{1.05}, \tag{3c}$$

where $RR_C$ is in mmol C m$^{-2}$ a$^{-1}$. $r_{C:P}$ (in Eq. 4) depends on the bottom water oxygen concentration and was calculated according to (Wallmann, 2010; equation 4).

$$r_{C:P} = Y_F - A \cdot \exp(-O_2/r). \tag{4}$$

where $O_2$ is in mmol m$^{-3}$ and the coefficients and their uncertainties are $Y_F$=123±24; $A$=112±24; $r$=32±19 mmol m$^{-3}$. Under high $O_2$ conditions $r_{C:P}$ is 123, which is close to the Redfield ratio of 106. Under low $O_2$ conditions, $r_{C:P}$ is lower than 106, which leads to a preferential P release from organic matter and, eventually, a net release of P from the sediment ($BEN_P > RR_P$, in Eq. 1).

Burial fluxes of P were applied in the simulations *Bur, Bur_Dun, Bur_low, Bur_high, Bur_noSG* and *Bur_res*. The default *Bur* model configuration uses Eq. (3) (Flögel et al., 2011) and the subgrid-scale bathymetry. Uncertainties in benthic P burial were examined by modifying this default model configuration.

In the *Bur_Dun* simulation (i.e. burial parameterization from Dunne et al. 2007), $BUR_C$ was calculated using Eq. (5) with $RR_C$ in mmol C m$^{-2}$ d$^{-1}$; Dunne et al. (2007):

$$BUR_C = RR_C \cdot \left[0.013 + \frac{0.53 \cdot RR_C^2}{(c + RR_C)^2}\right]. \tag{5}$$

Where $c$ = 7 mmol C m$^{-2}$ d$^{-1}$. This parameterization leads to high (low) organic C burial rates for high (low) organic C rain rates. This formulation is different to the standard formulation of burial in Eq. (3b, c) where burial depends on the C rain rates and in addition on the water depth. In the standard formulation, C burial is by definition one magnitude larger in slope and shelf regions compared to the deep ocean (see Eq. 3b, c).

We examined the sensitivity of P burial to the uncertainty of the parameters in Eq. (4) describing the carbon to phosphorus regeneration ratio $r_{C:P}$. Given means and standard deviations for the parameters $Y_F$=123±24; $A$=112±24; $r$=32±19 and assuming a Gaussian distribution, 100,000 independent coefficient combinations were assembled to calculate offline a range of global P burial estimates. For the offline calculation, preindustrial fields of $O_2$ and $RR_C$ were extracted from the simulation *Bur* with a temporal resolution fine enough to resolve seasonal variations in the data. Global P burial varied between 0.21 TmolP a$^{-1}$ (*Bur_low*) and 0.60 TmolP a$^{-1}$ (*Bur_high)* for a confidence interval of 90% (coefficients are shown in Table 1). Individual spin-ups were performed for the *Bur_low* and *Bur_high* simulation to check that the offline calculated P burial corresponded to the online values from the spin-up. Only minor differences between the $O_2$ fields of the *Bur* spin-up and the spin-ups for *Bur_low* and

*Bur_high* simulations were noted (not shown), which implies negligible errors in the offline calculation of the preindustrial global P burial.

For the simulation *Bur_noSG* (i.e. without subgrid-scale parameterization), P fluxes at the sediment-ocean interface were calculated using the coarser standard model bathymetry, which barely reproduce the global coverage of shelf areas (compare hypsometries in suppl. Fig. S1). This does not affect other processes like circulation, advection or mixing.

The implemented transfer functions (Eq. 2 and 4) assume unlimited local reservoirs of sedimentary P, meaning that the cumulative release of P may exceed the local inventory of P in the sediment if the benthic release is sustained over a longer period of time. In the simulation *Bur_res* (i.e. restricted release or P reservoir) we tested the impact of this simplification by applying sediment inventory restrictions to sediment P release. In accordance to Flögel et al. (2011), release of P from the deeper ocean (>1000 m) cannot exceed the rain rate of organic P to the sediment. For the continental shelf and slope, an upper limit sediment P inventory was calculated based on the following assumptions. We assume that the top 10 cm of the sediment column are mixed by organisms and are hence regarded as the active surface layer that is in contact with the overlying bottom water. Considering a mean porosity of 0.8 and a mean density of dry particles of 2.5 g cm$^{-3}$, the mass of solids in this layer is 5 g cm$^{-2}$ (Burwicz et al., 2011). The mean concentration of total P in continental shelf and slope sediments is 0.07 wt-% equal to 22.6 µmol/g (Baturin, 2007). Together, these assumptions convert to a maximum local inventory of total solid P in the active surface layer of $RES_{P,max}$ = 113 µmol cm$^{-2}$ (Eq. 6a). We assume that shelf and slope sediments can release up to 100 % of the total solid P under low oxygen conditions. The local P inventory ($RES_P$) can be fully replenished by P supply from the water column and any excess P is assumed to be permanently buried:

$$\{RES_P \in \mathbb{R} \mid 0 \geq RES_P \geq RES_{P,max}\} \tag{6a}$$

$$\frac{\Delta RES_P}{\Delta t} = RR_P - BEN_P \tag{6b}$$

Local values of $RES_P$ adjust during the spin-up according to the environmental conditions. Our pragmatic sediment inventory approach most likely overestimates the upper limit of P that can be released from the sediments. For example, under low $O_2$ conditions, part of the releasable or reactive P is transformed into authigenic P and permanently buried (Filippelli, 2001).

All *Bur* experiments applied a constant global weathering flux ($W_{P,const}$) as established during the respective spin-up run (see Table 1 for values of $W_{P,const}$ for the different *Bur* experiments).

$$W_P = W_{P,const} \tag{7}$$

## 2.4 Weathering Experiments

Uncertainties in the ocean P inventory due to weathering processes and anthropogenic fluxes of P were examined with the model simulations *Anthr, Weath0.05, Weath0.10, Weath0.15* and *Weath0.38*.

In simulations *Weath0.05, Weath0.10, Weath0.15, Weath0.38* (i.e. the number represents the preindustrial weathering flux) the global weathering flux of P to the ocean ($W_P$) was parameterized in terms of an anomaly relative to a preindustrial P weathering flux ($W_{P,0}$) according to Eq. (8).

$$W_P = W_{P,0} \cdot (f(NPP, SAT) - 1). \tag{8}$$

The weathering function $f$ is given in Eq. (9). Values of $W_{P,0}$ are given in Table 1 and derived below. The chosen anomaly approach assumes that, at steady state, $W_{P,0}$ is balanced by a respective global burial flux and hence can be neglected during the spin-up. In these simulations no benthic P burial was applied and for preindustrial conditions the weathering function $f(NPP,SAT)$ equals 1 and hence $W_P$ equals 0 TmolP a$^{-1}$. The dynamic weathering function $f$ (Eq. 9) was adopted from Niemeyer et al. (2017) and is originally based on an equation from Lenton and Britton (2006) for carbonate and silicate weathering. Following Niemeyer et al. (2017), we assumed that the release of P is proportional to the chemical weathering of silicates and carbonates on a global scale. Equation (9) describes the sensitivity of terrestrial weathering to the change of global terrestrial net primary production ($NPP$) and global mean surface air temperature ($SAT$):

$$f = 0.25 + 0.75 \cdot (NPP/NPP_0) \cdot \left(1 + 0.087(SAT - SAT_0)\right). \tag{9}$$

with $NPP_0$ and $SAT_0$ being the respective preindustrial values. Increasing $SAT$ and $NPP$ lead to enhanced weathering. The upper estimate of $W_{P,0}$ in *Weath0.38* was inferred from the P burial reference simulation *Bur*, assuming that the global integral of burial is compensated by the preindustrial global weathering flux (i.e. the global marine P inventory is in steady state). With the simulations *Weath0.05*, *Weath0.10*, *Weath0.15*, *Weath0.38* we explored the range of $W_{P,0}$ estimates as derived from observational studies, which range from 0.05 to 0.30 TmolP a$^{-1}$ (see Fig. 1, Benitez-Nelson, 2000; Compton et al., 2000; Ruttenberg, 2003). These studies give a range of total P fluxes to the oceans, which are higher than interfered from dissolved inorganic P fluxes shown already in previous studies (e.g. Martin and Meybeck, 1979; Rao and Berner, 1993) and in the Global News Model (Seitzinger et al., 2005). A small amount of fluvial P is delivered to the ocean as dissolved inorganic P, but the majority (90%) is particulate (inorganic and organic) P (Compton et al., 2000). The fast transformations between dissolved and particulate P in rivers (seconds to hours) (Withers and Jarvie, 2008) suggest a much higher amount of P that is available for marine organism than derived from dissolved inorganic P concentrations. A large amount of bioavailable P in rivers is present as loosely sorbed and iron-bound P. Estimates of bioavailable P are given in Fig. 1 (Benitez-Nelson, 2000; Compton et al., 2000; Ruttenberg, 2003), which are much higher than the estimates for dissolved inorganic P (0.018 TmolP a$^{-1}$ from Seitzinger et al. (2005) or 0.03 TmolP a$^{-1}$ from Filippelli (2002)). Taking into account only fluxes of dissolved inorganic P would strongly underestimate the effect of weathering fluxes as a P source to the ocean. The weathering parametrization (Eq. 9) was used to scale preindustrial fluvial fluxes of bioavailable P that is delivered in UVic to the ocean as dissolved inorganic P. In the model, no distinction was made between particular and dissolved fluvial fluxes of P. Uncertainties to other weathering parameterizations were not investigated in this study. Our parameterization predicts similar weathering rates to other weathering formulations (Meissner et al., 2012, their Fig. 6a). Since weathering is calculated on a global scale, we cannot study the effects of regional lithology and soil shielding on weathered P (Hartmann et al., 2014). UVic neither resolves the P cycle in the rivers, which is an active field for scientific research (Beusen et al., 2016; Harrison et al., 2019).

Finally, global anthropogenic P fluxes from fertilization, soil loss due to deforestation and sewage as projected by Filippelli (2008) were prescribed in the simulation *Anthr* (anthropogenic).

## 3. Uncertainties in Phosphorus Inventory

The large range of projected global phosphorus (P) fluxes to the ocean from sediments or weathering (Fig. 2a) leads to uncertainties in future P inventories by up to 60% of the present-day value until the year 5000 (Fig. 2b). All simulations show negligible differences in atmospheric $CO_2$ concentrations and hence undergo a similar climate development. Maximum $CO_2$ concentrations of 2200 ppmv were reached in the year 2250 and then declined to 1100 ppmv by the year 5000, comparable to results from Clark et al. (2016).

### 3.1. Fluvial P Fluxes: Weathering and Anthropogenic

Largest uncertainties in the P inventory are related to the large range of P weathering fluxes (Fig. 2, blue curves). Upper and lower estimates of P weathering fluxes differ by a factor of 6 (Fig. 2a, blue lines). In our weathering simulations, weathering anomalies depend linearly on the preindustrial weathering flux, $W_{P,0}$, estimate (see Eq. 8) because the climate development is essentially equal across the simulations. Therefore, the choice of $W_{P,0}$ (Fig. 1a) is a major source of uncertainty for projected future land-ocean P fluxes.

Weathering fluxes increased from the pre-industrial value by a factor of 2.5 until the year 5000 for atmospheric $CO_2$ concentrations of 1100 ppmv. This is comparable with the two- to four–fold increase in weathering fluxes estimated during OAE 2 approximately 91 Ma ago (Pogge von Strandmann et al., 2013) when atmospheric $CO_2$ concentrations increased to about 1000 ppmv (Damsté et al., 2008).

In contrast to weathering-induced P input, anthropogenic P fluxes (Filippelli, 2008) influence the global marine P inventory only in the near future (Fig. 2a, black dashed line). A decline in anthropogenic P fluxes after the year 2100 is expected due to the depletion of the easily reachable phosphorite mining reserves (Filippelli, 2008).

### 3.2. Sediment Fluxes: Parameterizations, Subgrid Bathymetry, Sediment Reservoir

The release of P from the sediment is strongly dependent on the $O_2$ concentration in the water above the sediments (Wallmann 2003; Flögel et al. 2011). Climate warming reduces $O_2$ solubility and ventilation of the ocean, which decreases the global $O_2$ content (more details in Sect. 4). The general decrease in ocean $O_2$ content may therefore cause preferential release of P from marine sediments. Differences in sediment P fluxes in our simulations are related to uncertainties in the parameterization of the transfer function (Fig. 2, red lines, -0.01 to 0.22 TmolP a$^{-1}$ by the year 5000), to different representations of the bathymetry (Fig. 2, purple dashed line, 0.06 (without subgrid) and 0.12 (*Bur*) TmolP a$^{-1}$) and to the way sediment P reservoirs in the sediment are represented (Fig. 2, purple solid line, -0.01 (limited reservoir) and 0.12 (unlimited reservoir, *Bur*) TmolP a$^{-1}$).

The global P burial of approximately 0.2 TmolP a$^{-1}$ (Fig. 3) (Filippelli and Delaney, 1996; Benitez-Nelson, 2000; Ruttenberg, 2003) is relatively well reproduced by simulations *Bur_low* and *Bur_Dun*. The simulation with the standard UVic bathymetry (*Bur_noSG*) underestimates P burial by 60% while the simulations *Bur_high*, *Bur* and *Bur_res* overestimate P burial by 180%, 90% and 80% with respect to estimates based on observations. The transient response of the P release to $O_2$ was stronger for simulations with low burial and vice versa (Fig. 2), except for simulation *Bur_res*. In *Bur_res*, a

significant reduction in the transient P release occurred due to the implementation of a finite P reservoir, with net global P loss due to enhanced burial at the end of the simulation. In the year 5000, global P concentrations increased in *Bur_res* by only 0.06 mmolP m$^{-3}$ compared to the global mean pre-industrial concentration of 2.17 mmolP m$^{-3}$. This is six-fold smaller than the increase of 0.36 mmolP m$^{-3}$ in simulation *Bur* with an assumed unlimited P reservoir. The small increase in the oceanic P inventory in *Bur_res* can be explained by the reduction in P sediment inventory rather than by changes in the rain rate of particulate organic matter to the sediment ($RR_C$). In *Bur*, a rapid increase in the benthic P release appeared in areas where the water turned suboxic and thus drove a positive benthic feedback between P release, productivity and deoxygenation (Fig. 2a). A limited supply of P from the sediment (*Bur_Res*) dampens this feedback.

Simulated pre-industrial $RR_C$ increased significantly from 180 to 1040 TgC a$^{-1}$ on the shelf and globally from 900 to 1500 TgC a$^{-1}$ compared to simulations without subgrid bathymetry. Pre-industrial $RR_C$ with subgrid bathymetry agrees better to estimates by Bohlen et al. (2012) (Table 2) and to other field data studies reporting a range from 900 to 2300 TgC a$^{-1}$ (Fig. 4) (Muller-Karger et al., 2005; Burdige, 2007; Dunne et al., 2007; Bohlen et al., 2012).

In summary, subgrid bathymetry leads to a substantial improvement of the representation of $RR_C$ to the sediment. More realistic benthic fluxes of P could be also attained by adjusting parameters for $r_{C:P}$ (Eq. 4) or by using the function of Dunne et al. (2007) to calculate $BUR_C$ (Eq. 5). The implementation of a finite P reservoir in the sediment has a substantial impact on the transient development of the global P inventory on millennial time scales. This is an important improvement relative to earlier work and should be considered in future studies.

## 4. Ocean Deoxygenation and Suboxia

Climate change influences ocean oxygen content by changes in circulation, ocean temperature and the degradation of organic matter. In warming surface waters, the solubility of O$_2$ decreases along with an increase in stratification, which together cause the deeper ocean to becomes less ventilated (Bopp et al., 2002; Matear and Hirst, 2003; Oschlies et al., 2018; Shaffer et al., 2009). Changes in export production and the degradation of organic matter in the ocean interior also affect O$_2$ content. In the following, we analyze the impact of different ocean P inventories on ocean deoxygenation and suboxia (Fig. 5). For a more detailed analysis we compare *Weath0.15* to the *Ref* simulation. In the *Weath0.15* simulation, the assumed preindustrial weathering flux compares well to estimates from observations (Fig. 1).

In the *Ref* simulation, global suboxic volume increased due to climate change from 0.3 to 1% until the year 5000 (similar to Schmittner et al., 2008) and the suboxic sediment area increased from 0.06 to 0.23% (Fig. 5, black line). In the *Weath0.15* simulation, the increase in suboxic volume (suboxic sediment area) was more than 2 (3) times higher than for the *Ref* simulation. The expansion of suboxic sediment areas was also enhanced for simulations with benthic fluxes, which could be related to regional feedbacks between increasing marine productivity, decreasing oxygen and enhanced sedimentary P release (Tsandev and Slomp, 2009). The explicitly simulated finite sedimentary P reservoir in simulation *Bur_res* places an upper limit to the benthic release of P and dampens these

regional feedbacks, resulting in a weaker spreading of suboxic waters by only 17% compared to the *Ref* simulation.

In the following sections, we show how the expansion of suboxia is related to net primary production in the ocean (NPP), the export of organic matter (Sect. 4.1) and to nitrogen limitation (Sect. 4.2).

Finally, we show how changes in $O_2$ solubility and utilization vary over time and affect the global $O_2$ inventory (Sect. 4.3). The latter approach gives another perspective because changes in $O_2$ inventories are a global integrated signal in comparison to the extent of suboxia, which are a consequence of more local processes.

### 4.1. Enhanced Biological Pump

The biological carbon pump can be summarized as the supply of biologically sequestered $CO_2$ to the deep ocean. In the euphotic zone phytoplankton and diazotrophs take up $CO_2$, a process that is intensified by elevated $PO_4$ concentrations in the surface ocean (Fig. 6a). Part of the organic matter sinks out of the euphotic zone (Fig. 6b) to the ocean interior, where it is respired using $O_2$. It is therefore P supply to the surface waters that explains the differences in deoxygenation between the

simulations. Circulation changes could also affect the supply of $O_2$ to the ocean interior. However, no significant differences in climate and circulation appeared among the simulations and therefore the global-warming induced circulation changes affected all simulations in the same way.

In the *Ref* simulation, net primary production (NPP, Fig. 6a black line) increased from 45 to 70 TmolP $a^{-1}$ (57 to 89 GtC $a^{-1}$) by the end of the simulation. In *Weath0.15*, enhanced P supply to the ocean led to

350 a doubling of NPP compared to the *Ref* simulation. The P inventory increased continuously, but NPP did not follow this trend and instead peaked in the year 4000. In the year 5000, all simulations, excluding *Weath0.38*, showed a similar response of NPP to the P addition with an increase in NPP of 19 TmolP $a^{-1}$ (relative to the *Ref* simulation) per 10% increase in P inventory. In *Weath0.38* the response was weaker and NPP increased by 8 TmolP $a^{-1}$ per 10% rise in the P concentration. P is less

effectively utilized in simulations with large oceanic P inventories. Higher ocean temperatures enhanced remineralization of organic matter in the shallower ocean so that the overall export to NPP ratio decreased from its preindustrial value of 0.12 to an average value among all simulations of 0.08 by the year 5000. This is because despite the warming-driven enhanced remineralization, the warming-driven intensification of ocean stratification leads to a decline in supply of nutrients to the surface layer

and reduced export production, in line with earlier studies (eg: Bopp et al., 2013, Landolfi et al., 2017). To summarize, NPP and export of organic matter is sensitive to P addition. However, the proposed positive feedback between P, NPP, export of organic matter, and deoxygenation was limited in our simulations due to a negative feedback related to nitrate availability. This is shown and explored in the following section. We acknowledge that accounting for P burial in weathering simulations may limit

the P increase. However, the effect of P burial has been shown to be small relative to the increase in benthic release of P due to the feedback involving redox-sensitive benthic P fluxes associated with the expansion of OMZ (Niemeyer et al. 2017, Fig. S1).

## 4.2. Nitrogen Limitation

At the end of the spin-up the N sink by denitrification and the N source by $N_2$-fixation were balanced. In the *Ref* simulation, climate warming enlarged the oxygen minimum zones, which enhanced denitrification in the tropics (not shown). In our model, diazotrophs are limited by P and Fe and are not limited by N. Their growth rate, which depends on temperature being zero below 15ºC, is slower relative to non-fixing phytoplankton. These characteristics allow them to succeed in warm, low-N and high–P environments that receive sufficient iron. In all simulations, $N_2$-fixation was stimulated by the addition of P to the ocean and was sensitive to rapid changes in the supply of P (compare Fig. 7a and Fig. 2a). However, $N_2$-fixers (Fig. 7a) were not able to use the extra P supply in polar and iron limited regions where low temperatures and iron limitation, respectively inhibit their growth (Fig. 8). This led to a substantial amount of excess phosphate in the surface waters of these regions (Fig. S2). Because $N_2$ fixers were not able to balance the loss by denitrification, nitrate decreased globally by 4 mmol N m$^{-3}$ by the year 5000 (Fig. 7b). The loss in nitrate led to a decrease in globally averaged N to P ratios. In the *Ref* simulation, N:P decreased from 14 to 12 and for the *Weath0.15* simulation it decreased to 10, which contributed further to a N limiting ocean. The nitrogen cycle was not able to recover from the decrease in N:P ratio with respect to pre-industrial values. We acknowledge that in the current study we did not account for potential future changes in iron concentrations (from atmospheric deposition, shelf inputs) and that the lack of a fully prognostic iron model may lead to a different sensitivity of the response of diazotrophs. Similarly, we did not account for the ability of phytoplankton to adapt to changing N:P ratios, that may affect marine biological productivity and in turn deoxygenation. These would require further studies.

## 4.3. Temporal Variations of Deoxygenation

Anomalies in circulation, ocean temperature and remineralisation of organic matter affect oceanic $O_2$ levels in a climate-warming scenario. In the *Ref* simulation, the $O_2$ inventory (Fig. 9a) decreased by 60 Pmol $O_2$ by the year 3000 and then reached present day values again by the year 5000. In *Weath0.15*, weathered P enhanced deoxygenation and led to a greater decrease in $O_2$ than in the *Ref* simulation. The $O_2$ decrease was up to 70 Pmol by the year 3300 and $O_2$ still showed a negative anomaly of 24 Pmol $O_2$ by the year 5000. Global anomalies in $O_2$ were due to changes of the Apparent Oxygen Utilization (AOU, Fig. 9b) and the $O_2$ saturation level (Fig. 9c). AOU is calculated from the difference between the $O_2$ saturation concentration and the in situ $O_2$ concentration assuming that all ocean water leaves the surface layer saturated in $O_2$. The calculation of AOU is in general biased to higher values, because in polar regions the water that is subducted and mixed into the deep water is undersaturated with respect to $O_2$ as a results of reduced air-sea gas transfer by sea ice (Ito et al., 2004). In UVic, this leads to an overestimation of AOU by 30% (Duteil et al., 2013). Sea ice cover reduces in a warming ocean that leads to an underestimation of the AOU anomaly in Fig. 9c. Changes in $O_2$ saturation were similar across the model simulations and lagged behinds surface ocean temperature changes. The circulation and ventilation of the ocean were similar in the model simulations because differences in surface temperatures were negligible and the atmospheric forcing of the ocean circulation was identical so that differences in AOU depended almost only on biological $O_2$ consumption and AOU anomalies

were directly yet inversely related to the changes in $O_2$ levels. Hence, biological consumption explained variations in $O_2$ content among the different model simulations (compare Fig. 9a and 9b). Increasing $O_2$ utilization contributed to the decrease of $O_2$ levels until the year 3000. Thereafter, a distinct negative trend in AOU with a similar slope was observed among all simulations and contributed to a re-oxygenation of the ocean. For simulations with larger P inventories, the AOU had a larger positive offset to the *Ref* simulation.

In a model with constant stoichiometry for elemental exchange by biological processes, anomalies in AOU (Fig. 10, blue lines) can be explained by the difference between total integrated nutrients (Fig. 10, red and black solid lines as anomalies) and preformed nutrients (Fig. 10, red and black dashed lines as anomalies). Preformed nutrients correspond to the fraction that leaves the surface ocean unutilized by phytoplankton (Ito and Follows, 2005). For example, in the Southern Ocean, a large fraction of nutrients that leaves the surface is preformed. The fraction of utilized and preformed nutrients can change during a transient simulation and could affect the oxygen state of the ocean.

In the *Ref* simulation (Fig. 10a), the anomaly of preformed dissolved inorganic P was directly inverse to the anomaly of AOU because the oceanic P Inventory was conserved in this simulation. Until the year 2200, changes in circulation and climate are the main cause for the reduction in preformed N and P in the *Ref* simulation since global N and P inventories were almost constant in this time period (Fig 9a, solid red and black line). During continuous and intense ocean warming, a weakening of the meridional overturning (not shown) reduced ocean ventilation. The meridional overturning maximum decreased from 17 Sv (pre-industrial) to 11 Sv in the year 2200. The continuous warming and stratification of the ocean reduces the supply of nutrients to the surface layer from the deep ocean. This is consistent with a reduction of the export of organic matter until the year 2200 (Fig. 6b). The balance between exported P out of the surface ocean and supplied P controls changes in AOU. We suggest that a weaker overturning increased the residence time of water and nutrients in the surface ocean. Nutrients staying longer in the euphotic zone are more likely to be biologically consumed. This implies more efficient utilization of nutrients and, hence, the reduction in preformed nutrients and an increase in AOU.

Enhanced suboxia after the year 2200 drove excess denitrification and a decline in nitrate (Fig. 10a red solid line) in the *Ref* simulation. The decline in nitrate could explain the negative trend in AOU anomalies (Fig. 10a blue solid line) and therefore a negative feedback on the global deoxygenation. In the year 2200, overturning had started to recover quickly and increased to 21 Sv in the year 3000 (+24% relative to preindustrial values), leading to faster overturning of organic matter in the surface ocean and a decrease in global AOU. This suggests that the slight increase in export by 5% (relative to preindustrial values) was not strong enough to compensate for the 24% faster overturning, which reduced the residence time of nutrients in the surface ocean.

P addition in the *Weath0.15* simulation stimulated $N_2$-fixation by diazotrophs and counteracted N-loss by denitrification (Fig. 10b, red solid line). This led to an increase in N inventory by 17 Pmol $O_2$-equivalents compared to the *Ref* simulation. Furthermore, the high availability of P seems to reduce preformed N by 6 Pmol $O_2$ equivalents. Both explain the difference in AOU between *Weath0.15* and *Ref* of 24 Pmol $O_2$ at the end of the simulation (Fig. 9b). However, denitrification still exceeded $N_2$-

fixation, which led to low levels of nitrate. From the year 5000 approximately all of the added P in the *Weath0.15* simulation remained unused by phytoplankton, left at the surface ocean as preformed P and was afterwards stored in the deep ocean. Phytoplankton was unable to utilize the extra P because it was limited by nitrate. Diazotrophs could not counteract the lack in N due to iron limitation and low surface temperatures in the polar oceans. The denitrification feedback driven by the spread of suboxic conditions in the tropics had reduced further the N availability for phytoplankton and limited the effect of P addition on the global oxygen level.

**5. Discussion and Conclusions**

In this study we compare simulations with different biogeochemical P settings but with virtually the same ocean circulation. We find that the $O_2$ and P inventories are very sensitive to the weathering and benthic P flux parameterizations tested in our model. Large uncertainties (Fig. 2, blue lines) derive from poorly constrained estimate for the preindustrial P weathering flux that ranges from 0.05 to 0.30 Tmol P $a^{-1}$ (Benitez-Nelson, 2000; Compton et al., 2000; Ruttenberg, 2003). The preindustrial weathering flux in simulation *Weath0.15* (0.15 Tmol P $a^{-1}$) is well in this range. In this simulation, enhanced weathering leads to an increase in the global ocean P inventory by 25% until the year 5000 (Fig. 2, blue dotted line). Benthic fluxes of P were simulated using transfer functions on a subgrid bathymetry. Applying the transfer functions without taking into account the local sedimentary P inventory can greatly overestimate the release of benthic P on long time scales. In the UVic model, the application of finite benthic P inventories limited the benthic release significantly. Under low-oxygen conditions, sediments were P depleted already after a few years to decades. In our simulation, this resulted in an increase in the global oceanic P inventory by just 3% (Fig. 2, magenta solid line). This could imply that benthic release of P is actually negligible in comparison to the weathering fluxes of P, but the UVic model does not resolve coastal processes such as the deposition of reactive particulate P from rivers on the continental shelves and its dissolution and release to the water column. For a more realistic comparison of benthic and fluvial P fluxes, a more detailed representation of coastal processes would be necessary to simulate deposition and release of fluvial P from the sediments at the shelf. However, we can conclude that the actual local inventories of P are too small to sustain a positive benthic P feedback over several millennial. Further, we find that a more realistic bathymetry substantially improves the simulated rain rate of particular organic carbon to the sediment (Table 2), particularly on the shelf, which most models do not resolve. Anthropogenic P fluxes increased the global P inventory by just 2% (Fig. 2, black dashed line). In summary, considering the weathering parameters closest to present day, the model formulation with limited P reservoir and anthropogenic fluxes from Filippelli (2008), assuming a linear combination of all P inputs, we find a +30% increase in the total global ocean P inventory by the year 5000. This seems to be surprisingly high, but several studies indicate that changes in past climate could also have been accompanied with substantial changes in the P inventory but at a much lower pace (Planavsky et al., 2010; Monteiro et al., 2012; Wallmann, 2014). In this simple addition of the P inventories we cannot account for feedbacks that might become apparent in a fully coupled model. For such high P inventories, we would expect larger

suboxia and therefore more P release from sediments and at the same time a stronger export of organic P and increased P burial.

The increased P inventory (Fig. 2b) promotes deoxygenation (Fig. 5) and expansion of suboxia, but it also causes a net loss of nitrate, which appears to further limit the full utilization of P by phytoplankton in our simulations. Wallmann (2003), using a box model, already recognized that for a eutrophic ocean,

nitrate might ultimately limit marine productivity. As a consequence, large amounts of P leave the surface ocean as preformed P (Fig. 10b) with no further impact on $O_2$ levels in the ocean interior. Low N/P ratios are thought to give $N_2$-fixers a competitive advantage over ordinary phytoplankton and lead to an increase in $N_2$-fixation (Fig. 7a). In the time period of the OAE1a and the OAE2, a substantial increase in $N_2$-fixation was also inferred from measurements of sediment nitrogen isotope

compositions typical for newly fixed nitrogen conditions and from high abundances of cyanobacteria indicated by a high 2-methylhopanoid index (Kuypers et al., 2004). However, high denitrification rates remove nitrate from the global ocean and in the UVic model $N_2$-fixers are not able to compensate for this loss (Fig. 7b) because low temperatures in polar regions and iron limitation at lower latitudes inhibit growth of diazotrophs (Fig. 8) and a substantial amount of excess phosphate remains in the

surface waters in these regions (Fig. S2). General circulation models without a N cycle, or box models without realistic representation of habitats suitable for $N_2$-fixers, would miss this important negative feedback limiting global deoxygenation. As a next step it would be reasonable to investigate how different parameterizations of the N cycle and a full dynamic iron cycle will affect the utilization of the added P. For example benthic denitrification is not simulated in the UVic model. Model simulations

showed for this century, that the enhanced denitrification in the water column could be compensated by less benthic denitrification (Landolfi et al., 2017), which could reduce the N-limitation and therefore enhance the effect of P fluxes on the biological pump. Sources of bioavailable Fe are still not well quantified and how these sources change under climate change is under debate (Hutchins et al., 2016; Mahowald et al., 2005). A more realistic representation of a dynamic iron cycle in UVic would affect

$N_2$-fixation in many areas of the global ocean (Fig. 8). Some additional model limitations are a cause for uncertainty in our results. We considered a fixed Redfield-ratio stoichiometry. In future deoxygenation studies, an optimality-based model for nutrient uptake with variable nutrient ratios (Pahlow et al., 2013) could be applied to investigate how well marine organisms adapt to a changing nutrient availability in the global ocean. Sea level change and the implied bathymetry change were not

simulated in the UVic model. In future projections, higher surface air temperatures would lead to a rise in sea level, which increase global coverage of shelf areas. Burial of P is more effective on the shelf (Flögel et al., 2011), which would remove P from the ocean and lead to a lower marine P residence time (Bjerrum et al., 2006). Finally, the model does not consider a fully prognostic (vertically-resolved) sediment model for C burial, which may reduce $O_2$ consumption in water depths shallower than 1k m.

To conclude, climate warming leads to a larger oceanic P inventory mainly due to addition of P by weathering, but also due to the release of P from the sediment and due to anthropogenic fluxes. A realistic representation of shelf bathymetry improves the predicted benthic P fluxes. Transfer functions for benthic P release should consider the sedimentary P inventory. However, the largest uncertainties in the projection of oceanic P inventory are due to poorly constrained weathering fluxes of P. Although

additional deoxygenation is driven by P addition to the ocean, the degree of deoxygenation – and hence the positive redox-related feedback on benthic P release is eventually limited by the availability of N and the apparent inability of the modelled $N_2$ fixation to respond to the larger P inventory.

**Acknowledgements.** This study is a contribution to the Sonderforschungsbereich (SFB) 754 "Climate-
530 Biogeochemical Interactions in the Tropical Ocean" and it was supported by the German Research Foundation through the Emmy Noether Program (independent junior research group ICONOX). We thank Wolfgang Koeve for his helpful and valuable comments.

**Data and Code Availability.** The model data and the model code are available at
535 https://data.geomar.de/thredds/catalog/open_access/kemena_et_al_2018_esd/catalog.html.

**Author contributions.** All authors discussed the results and wrote the manuscript. T.P.K. led the writing of the manuscript and the data analysis.

**Competing interests.** The authors declare that they have no conflict of interest.

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

**Tables:**

**Table 1: Overview of simulations. P fluxes are given in TmolP a$^{-1}$. We divided all simulations in four groups indicated by different colors. These are: reference simulations (in black) with and without anthropogenic fluxes of P; simulations with different formulations for the burial (in red beginning with the acronym Bur); simulations with weathering fluxes of P for different climate sensitivities (in blue beginning with the acronym Weath); and simulations with different representations of the sediment (in purple). In the P weathering simulations, only weathering anomalies were applied. The weathering flux in simulation *Anthr* is variable over time (Fig. 2a). In the P burial simulations, a constant P weathering flux ($W_{P,0}$) balances P burial ($BUR_P$) during the spin-up simulations. The preindustrial P inventory is identical in all simulations. More detailed information can be found in the text.**

| Simulations | Abbreviation | Fluxes | P Burial parametrization |
|---|---|---|---|
| Reference (constant P inv.) | *Ref* | No | No burial |
| Anthropogenic P input | *Anthr* | Flux from Filippelli (2008) | No burial |
| Burial Reference | *Bur* | $BUR_P$ (t=1775a)=0.38 $W_{P,const}$=0.38 | $r_{C:P}$ (Wallmann, 2010), C Burial (Flögel et al., 2011) $Y_F$=123; $A$=112; $r$=32 in Eq. 4 |
| Burial Dunne | *Bur_Dun* | $BUR_P$ (t=1775a)=0.25 $W_{P,const}$=0.25 | $r_{C:P}$ (Wallmann, 2010), C Burial (Dunne et al., 2007) |
| Low burial estimate | *Bur_low* | $BUR_P$ (t=1775a)=0.21 $W_{P,const}$=0.21 | *Bur* configuration, but with $Y_F$=100.5; $A$=90; $r$=38 in Eq. 4 |
| High burial estimate | *Bur_high* | $BUR_P$ (t=1775a)=0.60 $W_{P,const}$=0.60 | *Bur* configuration, but with $Y_F$=167; $A$=108.5; $r$=29.5 in Eq. 4 |
| Burial without subgrid bathymetry | *Bur_noSG* | $BUR_P$ (t=1775a)=0.09 $W_{P,const}$=0.09 | *Bur* configuration, but without subgrid bathymetry |
| Burial with restricted reservoir | *Bur_res* | $BUR_P$ (t=1775a)=0.41 $W_{P,const}$=0.41 | *Bur* configuration, but with 113 μmolP cm$^{-2}$ Reservoir |
| Weathering | *Weath0.05* | $W_{P,0}$=0.05 | *No burial* |
| Weathering | *Weath0.10* | $W_{P,0}$=0.10 | *No burial* |
| Weathering | *Weath0.15* | $W_{P,0}$=0.15 | *No burial* |
| Weathering | *Weath0.38* | $W_{P,0}$=0.38 | *No burial* |

**Table 2: Rain rate of particulate organic carbon ($RR_C$) to the seafloor for the shelf, slope and deep-sea areas from the observational estimate by Bohlen et al. (2012) and for the UVic model simulation *Bur* with and without subgrid bathymetry. Preindustrial RR$_C$ shows no significant differences among all model simulations (expect for simulation *Bur_noSG*).**

| | Depth [m] | Bohlen (2012) | | | UVic model with subgrid bath. (Simulation *Bur*) | | | UVic model without subgrid bath. (Simulation *Bur_noSG*) | | |
|---|---|---|---|---|---|---|---|---|---|---|
| | | $RR_C$ [TgC a$^{-1}$] | $RR_C$ [%] | Area [%] | $RR_C$ [TgC a$^{-1}$] | $RR_C$ [%] | Area [%] | $RR_C$ [TgC a$^{-1}$] | $RR_C$ [%] | Area [%] |
| **Shelf** | 0-200 | 1056 | 60 | 6 | 1039 | 70 | 6.5 | 179 | 28 | 2.3 |
| **Slope** | 200-2000 | 393 | 22 | 10 | 205 | 14 | 11.7 | 219 | 34 | 13.3 |
| **Deep sea** | >2000 | 312 | 18 | 84 | 235 | 16 | 81.9 | 238 | 37 | 84.6 |
| **Sum** | | 1761 | | | 1479 | | | 637 | | |

**Figures:**

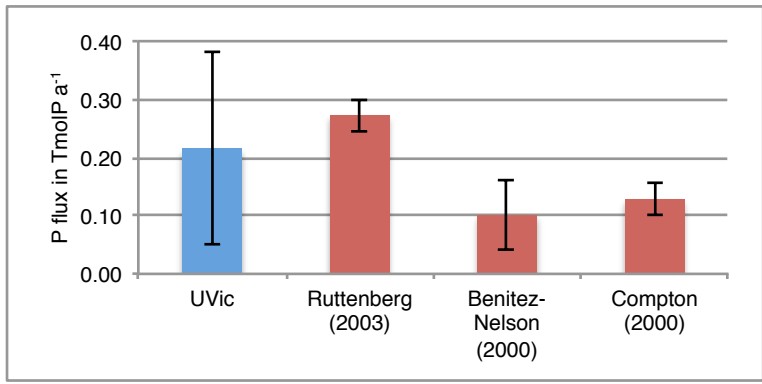

**Fig. 1: Globally integrated preindustrial P weathering fluxes in TmolP a⁻¹ from field studies(red) and the range of preindustrial P weathering fluxes covered by all simulations (blue with bars indicating the range; see $W_{P,0}$ in Table 1). Estimates from field studies are based on literature values for global fluvial fluxes of bioavailable P and the error bars denote upper and lower limits of these estimates.**

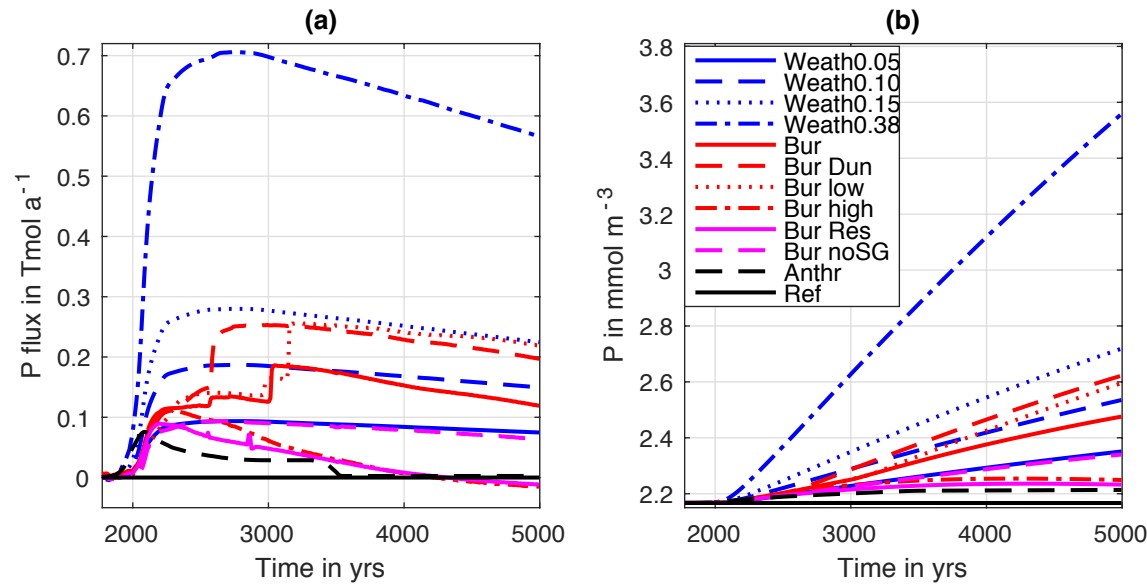

**Fig. 2: (a) Globally integrated flux of P in Tmol a⁻¹ to the ocean and (b) globally averaged phosphate concentration in mmol m⁻³. Simulation descriptions can be found in Table 1.**

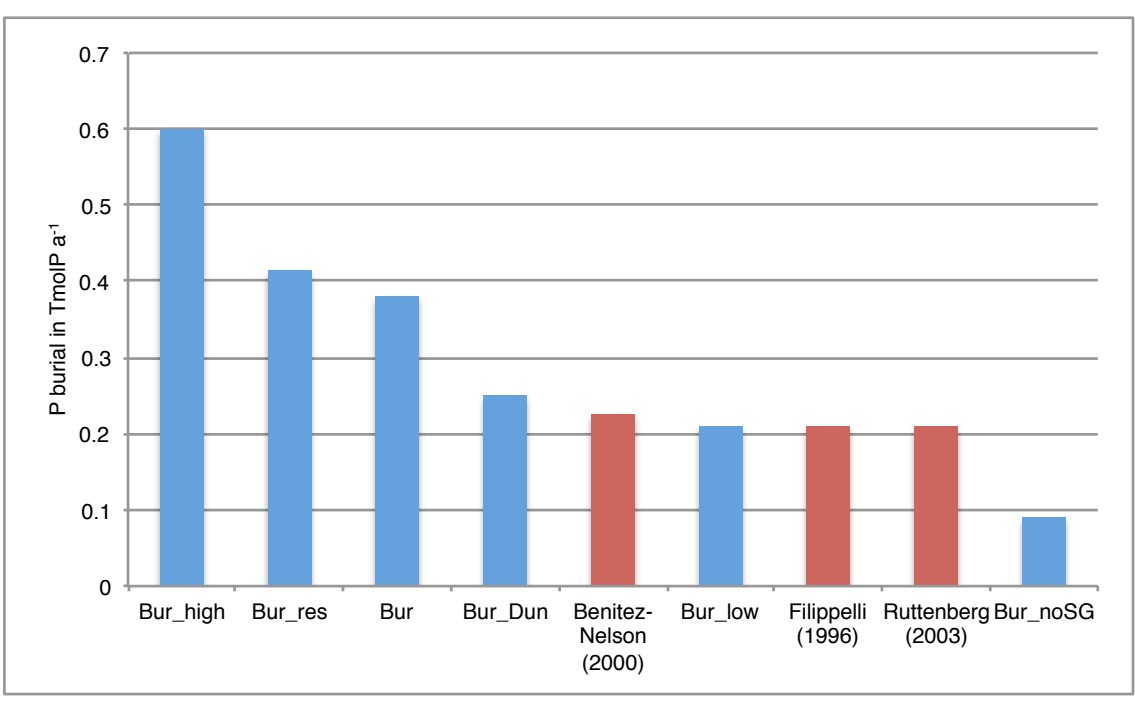

**Fig. 3: Globally integrated preindustrial P burial fluxes in TmolP a⁻¹ from field studies (red) and for UVic model simulations in the year 1775 (blue). Description of the model simulations can be found in Table 1.**

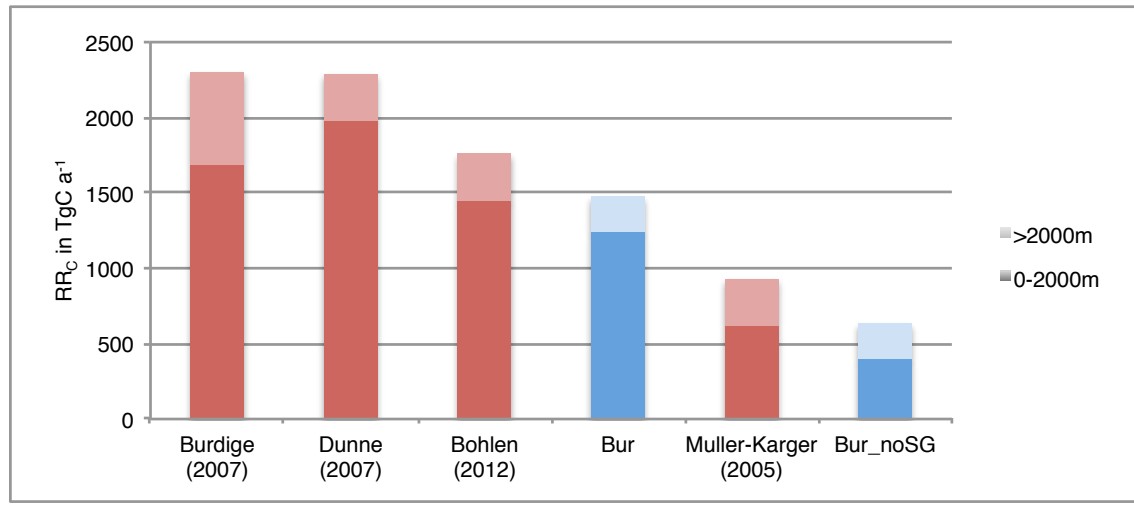

**Fig. 4: Globally integrated preindustrial rain rate of particulate organic carbon ($RR_C$) to the seafloor in TmolC a⁻¹ from published studies (red) and for UVic model simulations (blue) between 0 to 2000m water depth (dark blue) and below 2000m (light blue). The simulation *Bur* is representative for all UVic model simulations except *Bur_noSG*.**

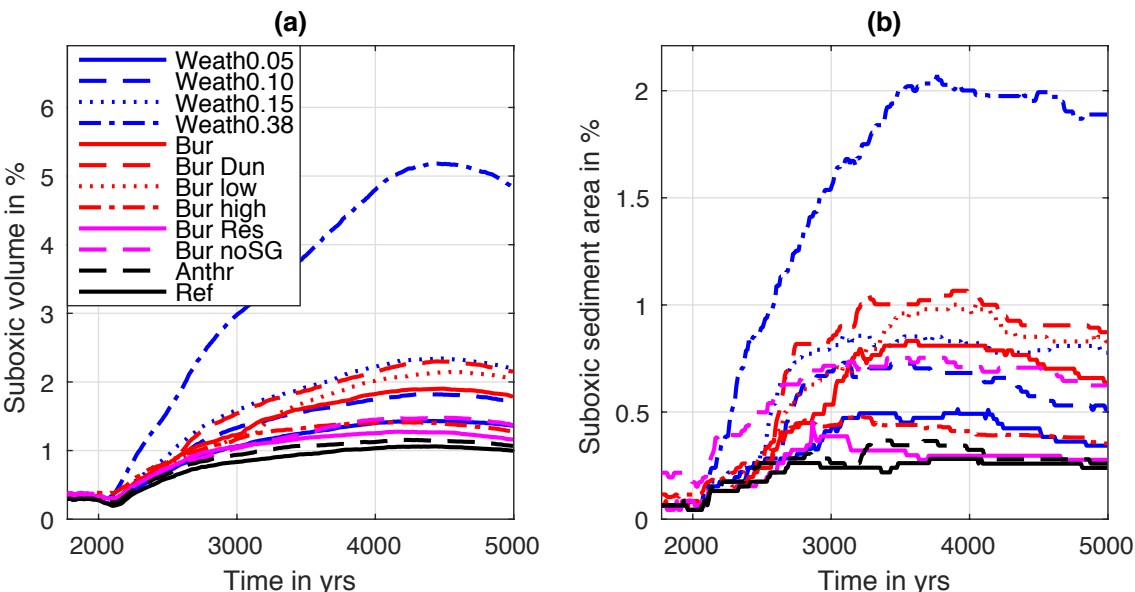

**Fig. 5: Globally integrated (a) suboxic volume in percentage of total ocean volume and (b) suboxic sediment surface area in percentage of total sediment surface area. Water is designated as suboxic for oxygen concentrations below 5 mmolO$_2$ m$^{-3}$. Simulation descriptions can be found in Table 1.**

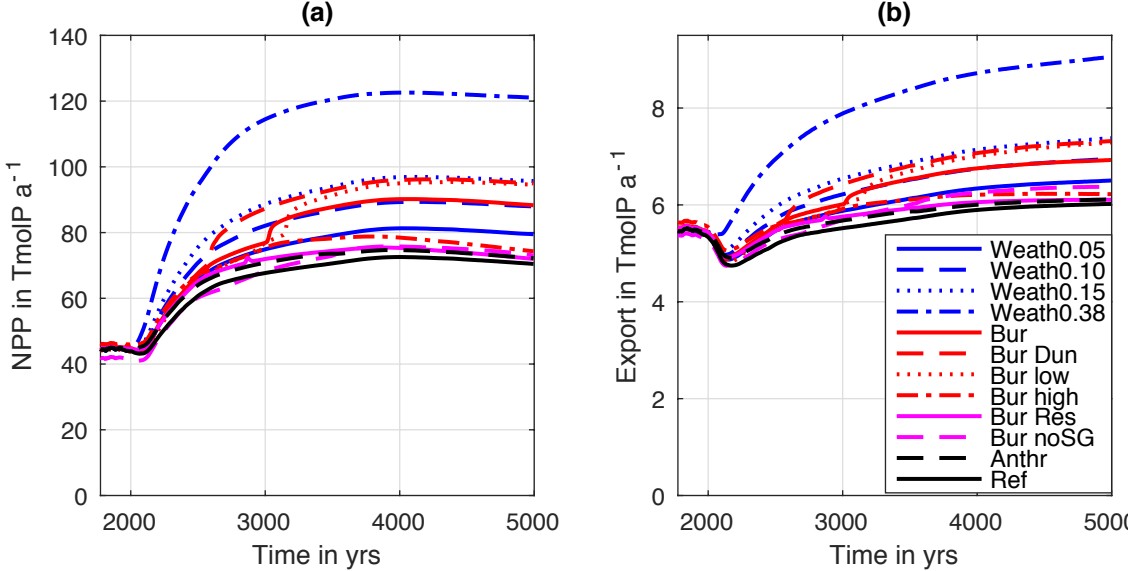

**Fig. 6: Globally integrated (a) ocean net primary production (NPP) in TmolP a$^{-1}$ and (b) export of organic P below the 130m depth level in TmolP a$^{-1}$. Simulation descriptions can be found in Table 1.**

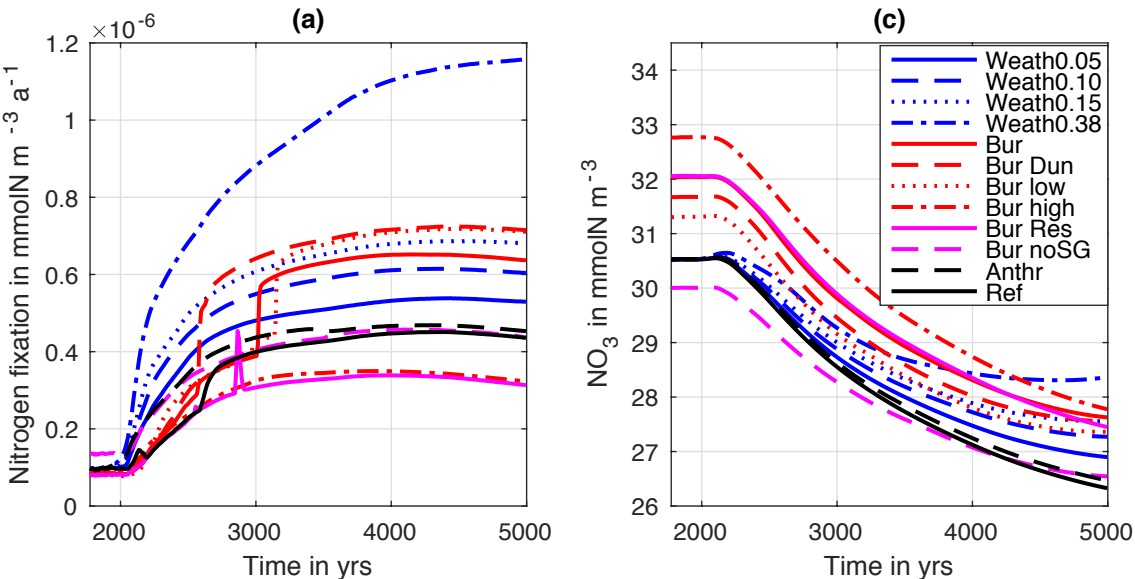

**Fig. 7: Globally averaged (a) N₂-fixation in mmolN m⁻³ a⁻¹ and (b) NO₃⁻ concentration in mmolN m⁻³. Simulation descriptions can be found in Table 1.**

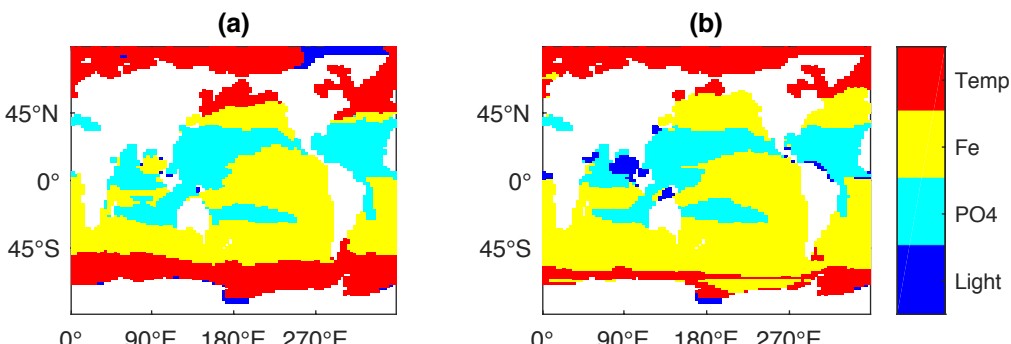

**Fig. 8: Spatial distribution of the most limiting factors for growth of diazotrophs for (a) the preindustrial case and (b) simulation year 5000 for *Weath0.15*. Limitation of iron (Fe) and phosphate (PO4) are based on Monod kinetics so that the limitation factors vary between 0 and 1. The light limitation factor also varies between 0 and 1. In the model, diazotrophs only grow at temperatures higher than 15.7 °C. For temperatures above 15.7 °C, diazotroph growth depends on the equation exp(T/15.7°C)-2.61. Diazotroph growth is not limited by nitrate availability in the model. A more detailed description of diazotroph growth and iron limitation can be found in Keller et al. (2012) and Nickelsen et al. (2015).**

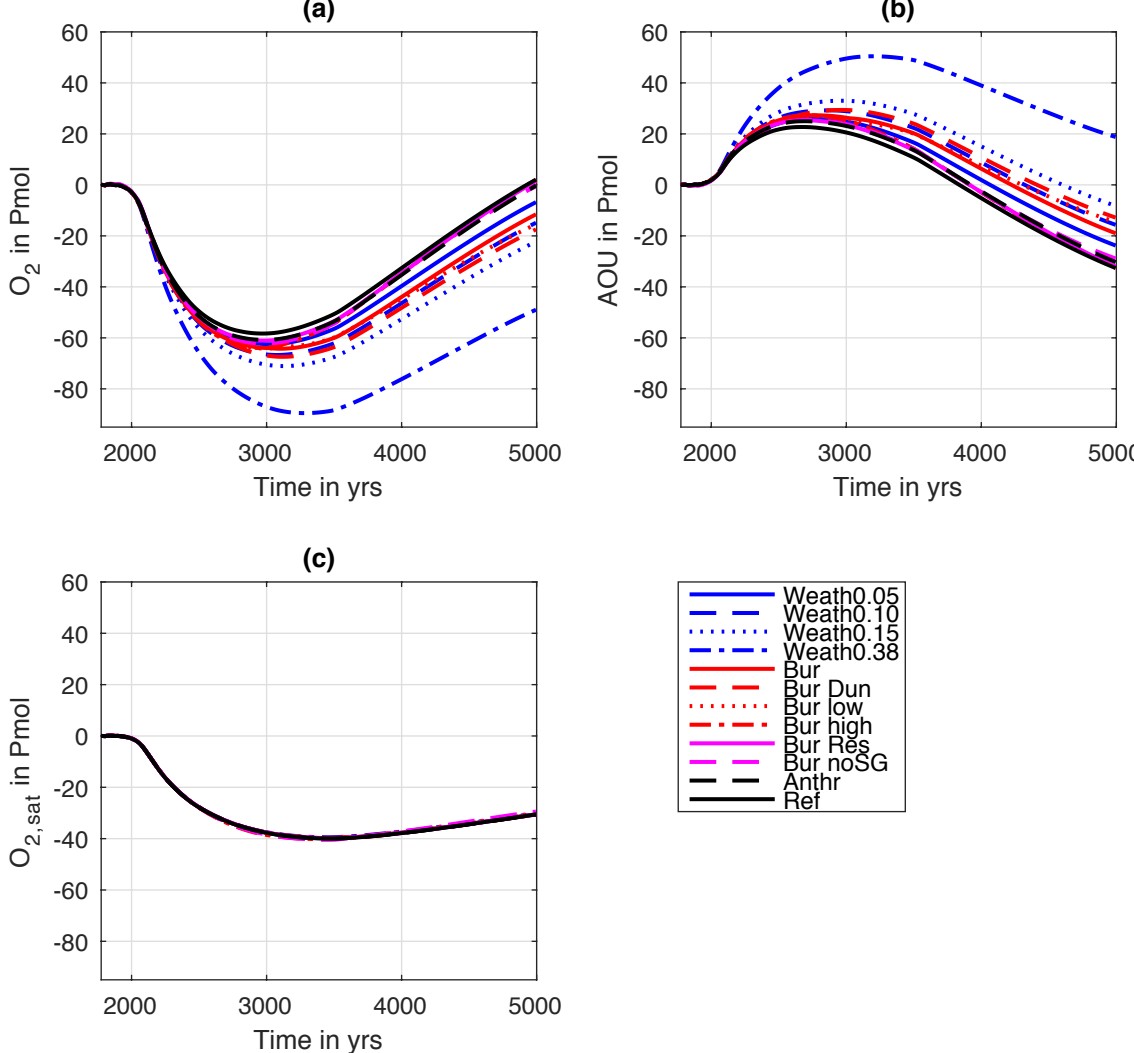

**Fig. 9: Anomalies of globally integrated (a) O$_2$ content, (b) apparent oxygen utilization (AOU) and (c) oxygen saturation (O$_{2,sat}$) in Pmol O$_2$. Simulation descriptions can be found in Table 1.**

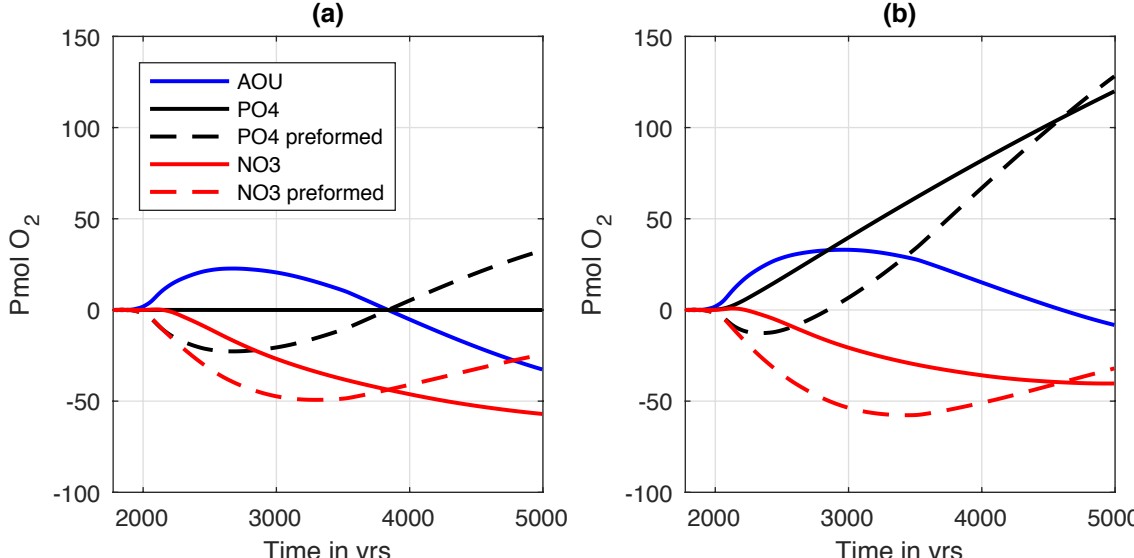

**Fig. 10: Anomalies of globally integrated AOU (blue line), PO₄³⁻ (black solid line), preformed PO₄³⁻ (black dashed line), NO₃⁻ (red solid line) and preformed NO₃⁻ (red dashed line) expressed in Pmol O₂ equivalents using constant elemental ratios (O:N=10 and O:P=160) for the (a) *Ref* simulation and the (b) *Weath0.15* simulation. Preformed nutrients are calculated as the difference between remineralized and total nutrient content. The calculations assume that all ocean water leave the surface layer saturated in O₂.**