# Peer review of "Fig. S1: Hypsometry for the upper ocean for the model bathymetry (blue and red lines) versus observations (black line) from ETOPO2v2 (National Geophysical Data Center, 2006)."

_Earth System Dynamics, 2018_

## Referee Comment (RC1) · Anonymous Referee #1 · 7 Dec 2018

This is an interesting paper describing potential future changes in marine phosphorus (P) cycling over the coming two millennia as obtained with an earth system model of intermediate complexity. The major conclusions are that there are large uncertainties in these projections due to our lack of knowledge of the expected changes in P supply due to weathering and benthic release. An interesting observation is that, in this model, nitrogen fixation cannot keep up with P supply. The paper is generally well-written and deserves rapid publication. I do have a number of recommendations that I suggest the authors consider in a revision:

1. The presentation of the scenarios could be improved. In the model, 12 different

runs were performed to explore how the marine P cycle responds to different model settings. The differences in the results of these scenarios play a crucial role but it takes the reader quite some time to figure out what is what. This could be improved if the sequence of the 12 scenarios in the text would be the same as in Table 1. A brief explanation of the various groups of scenarios could also be added in the caption of figure 2.

2. The reasons for the N limitation upon ocean deoxygenation could be mentioned explicitly in the abstract.

3. It would be helpful to the reader to explicitly discuss the model assumptions leading to N limitation including uncertainties in changes in Fe cycling. I also would suggest to move Fig. S2 to the main paper.

4. There is a lot of recent work on river fluxes of P to the ocean that would be appropriate to reference for context.

Detailed comments (partly overlapping): - Line 10: suggested change: "….that enhanced weathering and increased benthic phosphorus (P) fluxes"

- Line 14 and 15 and elsewhere: "until the year 2300"

- Line 25: suggested change: "In the model, nitrogen fixation was not able to adjust…":

- Line 25. Here, it would help if the authors clarify why nitrogen fixation does not adjust to the high P levels. Because these are model results, this can be specified.

- Line 27: "this contrasts with"

- Line 27. Here, the authors could clarify whether the palaeo reconstructions refer to model studies or reconstructions based on data or both and exactly how those results are different.

- Line 49. Suggested change "the Earth has experienced" or "the Earth experienced"

[Figure]

- Line 49. It's not clear what is meant by "climate OAE-like states". I would suggest to rephrase.

- Line 66. What about increased inputs of Fe from continental shelves upon ocean deoxygenation? Could they alleviate the Fe limitation in the model? What is the uncertainty in the dust inputs?

- Line 79. "and improved representations of"

- Line 96: "as the switching point"

- Line 98: "so that" (instead of "such that")

- Line 111 and Table 1: are benthic P fluxes equated to burial? I don't see the term benthic flux in Table 1. The terms need to be used consistently.

- Line 112: in a print, the purple and blue are very difficult to differentiate

- Table 1: please add much more detail on the abbreviations in the text (or can you think of an easier notation?). Now it is very hard for the reader to keep the various model scenarios apart. Note that the sequence doesn't match the text and that the anthropogenic flux not quantified.

- Line 136-138. Please explain this section on the organic C burial better: it seems contradictory that all organic C is remineralized but that there is still organic C burial.

- Line 258. "is essentially equal"

- Lines 225-226. It is very well known already for a long time that total P in rivers can be mobilized in the coastal zone and forms a key input of P to the coastal zone. These are not new findings of Benitez-Nelson, Compton et al. and Ruttenberg as suggested here, so I would suggest to rephrase this sentence.

- Same section: you could consider including a reference to the weathering & anthropogenic flux of P calculated from the Global News models (see the work of Seitzinger

et al. 2010; Global Biogeochemical Cycles) and follow-up studies (e.g. Beusen et al. 2016; Biogeosciences)

- Lines 258. Change to "essentially"

- Line 267. Change to "after the year" (note that "the" is missing before year in more places)

- Line 327: Change to "a consequence"

- Line 350. Change to "shown"

- Line 358-359. This is an important piece of information that also should be given in the abstract, see comment above.

- Don't start a sentence with "So that"

- Line 440-443. Needs rephrasing since benthic P release is known to be important in the coastal zone from both experimental field studies and modeling.

- Line 459. The term "palaeo study" is vague. Please provide more information i.e. on the type of setting and time period.

- Line 460. "not able to compensate. . .." Unless you are underestimating the Fe input to the ocean. A few lines on the uncertainties there (e.g. shelf Fe input, dustfield) would be useful.

- 'line 480. Could be changed to "on benthic P fluxes in this model is eventually. . ." These results really depend on how the N and Fe cycles are parameterized – should be discussed in a few more sentences in the main text.

- Fig. S2: I would suggest to move this figure to the main paper.

---

## Referee Comment (RC2) · Anonymous Referee #2 · 21 Dec 2018

Kemena et al. present 12 long-term global warming simulations of the UVic Earth System Model of Intermediate Complexity to assess how projected P and O2 inventories depend on implemented weathering and sedimentary fluxes. The focus is on biogeochemical feedbacks, as the physical response is almost identical across the simulations. They suggest weathering fluxes contribute most to projected increased P inventories. I consider this to be a welcome contribution to the field of long-term Earth system projections. I have several questions regarding the methodological approach and conclusions which should be considered to improve the readability and focus of the manuscript.

[Figure]

-Bur simulations: I find it hard to judge the "meaningfulness" of the Bur simulations given that they release P from an unlimited reservoir. The Bur_Res simulation seems to indicate an upper limit on the potential P release, which all other Bur simulations appear to surpass. So all other simulations release more P than can be assumed to be in the sediments, correct? Should the reservoir constraint not apply to all simulations? You might want to consider disqualifying those simulations a bit faster in the text and highlight the importance of this constraint a bit clearer, including the abstract and method/results section.

-Weathering simulations: Why do these simulations not have a burial formulation? Of course, addition of P to an otherwise "closed" ocean just increase its P inventory. Would it not be important to assess the feedbacks associated with deposition and redissolution in this context, as increased P supply to the upper ocean stimulates NPP, export and deposition? I could not find a discussion on this matter in the manuscript.

-It is not clear to me which simulation corresponds to the best estimate reported in the abstract and conclusion. Do you just add the result of the Weath0.15, Bur_Res and Anthr simulations? You may want to guide the reader a bit better here, and explain why these simulations are simply additive? That seems strange to me, as for instance, the Bur_Res simulation would suggest a W0=0.41 TmolPa-1, much higher than the 0.15 TmolPa-1 considered in Weath0.15.

-Language: I find the language at times ambiguous. Most importantly, the usage of benthic fluxes, burial and sediment release appear confusing, and it's not clear whether gross or net exchange is meant. Please consider explicitly introducing these terms and using them consistently.

-Balance between presentation of P and O2 response: Even though deoxygenation is mentioned in the title, there is very limited mentioning of deoxygenation in the abstract and discussion. The most important driver of ocean deoxygenation appear to still be circulation changes, and the assessed biogeochemical feedbacks should be presented

in this context. There are several papers worth citing/discussing in the introduction and discussion on the matter of long-term projections of ocean deoxygenation, e.g.: Battaglia & Joss 2018, ESD, Yamamoto et al. 2015, GBC, Schmittner et al. 2008, GBC As such, the modeled circulation response may be compared to other long-term projections.

Specific comments:

Line 27: "this is in contrast to paleo reconstructions": not clear what is meant from the text

Line 29: "more reliable projections of ocean deoxygenation": context of ocean deoxygenation does not emerge from the abstract. How do the biogeochemical feedbacks assessed compare to uncertainties in circulation changes?

Introduction: centers round CO2-driven ocean deoxygenation. Please include recent literature on ocean deoxygenation, and the fact that circulation changes are crucial for ocean deoxygenation associated with global warming, not CO2 per se.

Line 57: "could": will?

2.3/2.4: Please consider presenting weathering first, consistent with presentation of results/Table 1

Line 126: "every grid box": every bottom grid box?

Line 137: "all organic C is remineralized in the deepest ocean layer": statement must be wrong?

Eq. 3a-c: potentially include z<1000m and z>1000m on the respective lines for clarity.

Eq. 4: Is there only a O2-feedback on P fluxes? Should the C burial/redissolution not also be O2-dependent? Potentially worthy of discussion.

Line 170: Please add numbers in parenthesis.

Line 183-200: Hard to understand. Examples below:

Line 187/188: "for the continental shelf and slope": how was this done for all other grid cells?

Line 194: "local inventory": what do you mean with this? Do all cells have this inventory? Or is this an upper limit for inventories globally?

Line 195/196: reservoir can be replenished, but excess P is permanently buried? Is the ∆RESp the replenished P or excess P?

Eq. 6: are the > and > correct?

Line 197: "depending on environmental conditions": what do you mean with this statement? Depending how? Earlier you mentioned local inventories of 113 umol/cm2?

Line 277: "the way sediment P reservoirs are represented": if represented at all or not.

3.2: Preferably start section with PI RR (lines 295-299) and PI burial rates (lines 279-283). Then, would strongly suggest changing the tone of this section, in that Bur simulations without a reservoir constraint are not realistic. Potentially exclude those runs from Figs.5-8, Figures are very busy anyway and lines are hard to tell apart. Or explain why those are considered for assessment of ocean deoxygenation, still.

Line 308/309: Please add more citations of long-term O2 projections.

Line 374: "recovered": strange language. O2 inventory is still increasing and simulation has not reached steady state yet.

Line 377: How is AOU/O2sat calculated? Potentially discuss/mention Ito et al., 2004, GRL?

Line 390: Are preformed nutrients carried as explicit tracers? Please introduce how signal is separated.

Line 397: "are likely": how come you are not sure about this?

Line 398: "global N inventory constant": does not appear to be the case in Fig. 9a?

Line 399: Physical response: potentially summarize physical response earlier, perhaps right after 4., as this is the baseline response of the scenario which applies to all variables?

Line 400: is this the max of the global meridional overturning?

Line 402: "consistent with reduction of export": in section 4.1 you appear to conclude that warmer temperatures enhance remineralization in the shallower ocean which reduces export?

Line 404-407: "speculate"? This reasoning, also in line 415, I do not understand. I would speculate that these changes are associated with older water masses. AOU and ideal age probably are highly correlated, indicating that more O2 is consumed in older waters. See literature.

Table 1: Bur_low and Bur_high: reference to equation (4) rather than (3)? Bur: potentially also add coefficients here for consistency with Bur_low, Bur_high

Figure 2a: net flux? What are the step-like increases in the Bur simulations associated with? Those are also present in subsequent figures?

Figure S2/S3: might benefit from an improved aspect ratio.

———————————————

---

## Author Comment (AC1) · 6 Feb 2019

First of all, we thank the reviewer very much for the constructive comments and advices, which helped to improve the manuscript a lot. Please find our reply below. Best regards,

Tronje Kemena and co-authors

Reviewer comments: This is an interesting paper describing potential future changes in marine phosphorus (P) cycling over the coming two millennia as obtained with an earth system model of intermediate complexity. The major conclusions are that there

are large uncertainties in these projections due to our lack of knowledge of the expected changes in P supply due to weathering and benthic release. An interesting observation is that, in this model, nitrogen fixation cannot keep up with P supply. The paper is generally well-written and deserves rapid publication. I do have a number of recommendations that I suggest the authors consider in a revision:

1. The presentation of the scenarios could be improved. In the model, 12 different runs were performed to explore how the marine P cycle responds to different model settings. The differences in the results of these scenarios play a crucial role but it takes the reader quite some time to figure out what is what. This could be improved if the sequence of the 12 scenarios in the text would be the same as in Table 1. A brief explanation of the various groups of scenarios could also be added in the caption of figure 2. We changed the sequence in Table 1 to fit the description in section 2 (with exception for the simulation Anthr). The descriptions and sequence of the burial experiments were also improved. We start in section 2.3 (burial experiments) with a general description how sinking organic matter interacts with the subgrid bathymetry and than explain how benthic fluxes of P are calculated: "The water column model is not coupled to a prognostic and vertically resolved sediment model. Instead, sinking organic matter interacts with the sediment via "transfer functions" (Wallmann, 2010) on a detailed subgrid bathymetry (Somes et al., 2013). Sinking organic matter is partially intercepted at the bottom of each grid box by a sediment layer and the intercepted amount depends linearly on the fractional coverage of the grid box by seafloor. The intercepted organic P is remineralized in accordance with Eq. (1) and Eq. (2), whereby organic C and N are completely remineralized under oxygen or nitrate utilization without any burial. Fractional coverage of every ocean grid box by seafloor was calculated on each model depth level according to the subgrid bathymetry (Somes et al., 2013). The subgrid bathymetry was inferred from ETOPO2v2 (National Geophysical Data Center, 2006). ETOPO2v2 has a horizontal resolution of 2-minutes fine enough to adequately represent continental shelves and slopes. The coarse standard model bathymetry in the UVic model has a horizontal resolution of 1.8° latitude x 3.6° longitude." (Line

126-137)

We added also a brief description of the different groups of scenarios to the caption of Table 1: "We divided all simulations in four groups indicated by different colors. These are: reference simulations (in black) with and without anthropogenic fluxes of P; simulations with different formulations for the burial (in red beginning with the acronym Bur); simulations with weathering fluxes of P for different climate sensitivities (in blue beginning with the acronym Weath); and simulations with different representations of the sediment (in purple)."

2. The reasons for the N limitation upon ocean deoxygenation could be mentioned explicitly in the abstract. We mention now the reasons for the N limitation already in the abstract: "In the model, nitrogen fixation was not able to adjust the oceanic nitrogen inventory to the increasing P levels or to compensate for the nitrogen loss due to increased denitrification. This is because low temperatures and iron limitation inhibited the uptake of the extra P and growth by nitrogen fixers in polar and lower latitude regions."(Line 25)

3. It would be helpful to the reader to explicitly discuss the model assumptions leading to N limitation including uncertainties in changes in Fe cycling. I also would suggest to move Fig. S2 to the main paper. We mention and discuss in section 4.2 model assumptions for N uptake of diazotrophs and we mentioned how Fe limitation is simulated: "In our model, diazotrophs are limited by P and Fe and are not limited by N. Their growth rate, which depends on temperature being zero below 15°C, is slower relative to non-fixing phytoplankton. These characteristics allow them to succeed in warm, low-N and high–P environments that receive sufficient iron."(Line 362) "We acknowledge that in the current study we did not account for potential future changes in iron concentrations (from atmospheric deposition, shelf inputs) and that the lack of a fully prognostic iron model may lead to a different sensitivity of the response of diazotrophs. Similarly we did not account for the ability of phytoplankton to adapt to changing N:P ratios, that may affect marine biological productivity and in turn deoxygenation. These would require

further studies."(Line 374)

Further, we discussed how N-limitation and possible future changes in the Fe-cycle could affect the utilization of P by diazotrophs to section 5: "As a next step it would be reasonable to investigate how different parameterizations of the N cycle and a full dynamic iron cycle will affect the utilization of the added P. For example benthic denitrification is not simulated in the UVic model. Model simulations showed for this century, that the enhanced denitrification in the water column could be compensated by less benthic denitrification (Landolfi et al., 2017), which could reduce the N-limitation and therefore enhance the effect of P fluxes on the biological pump. Sources of bioavailable Fe are still not well quantified and how these sources change under climate change is under debate (Hutchins et al., 2016; Mahowald et al., 2005). A more realistic representation of a dynamic iron cycle in UVic would affect N2-fixation in many areas of the global ocean (Fig. 8)." (Line 490)

We moved Fig. S2 to the manuscript and adapted the numbering.

4. There is a lot of recent work on river fluxes of P to the ocean that would be appropriate to reference for context. We referenced to following publication in the text (Line 250, 234, 241): Harrison, J. A., Beusen, A. H., Fink, G., Tang, T., Strokal, M., Bouwman, A. F., Metson, G. S., and Vilmin, L.: Modeling phosphorus in rivers at the global scale: recent successes, remaining challenges, and near-term opportunities, Curr. Opin. Environ. Sustain., 36, 68–77, doi: 10.1016/j.cosust.2018.10.010, 2019. Beusen, A. H. W., Bouwman, A. F., Van Beek, L. P. H., Mogollón, J. M., and Middelburg, J. J.: Global riverine N and P transport to ocean increased during the 20th century despite increased retention along the aquatic continuum, Biogeosciences, 13, 2441–2451, doi: 10.5194/bg-13-2441-2016, 2016. Seitzinger, S. P., Harrison, J. A., Dumont, E., Beusen, A. H. W., and Bouwman, A. F.: Sources and delivery of carbon, nitrogen, and phosphorus to the coastal zone: An overview of Global Nutrient Export from Watersheds (NEWS) models and their application, Global Biogeochem. Cycles, 19, 1–11, doi: 10.1029/2005GB002606, 2005.

Detailed comments (partly overlapping): - Line 10: suggested change: ": : :.that enhanced weathering and increased benthic phosphorus (P) fluxes" Added "that".

- Line 14 and 15 and elsewhere: "until the year 2300" Corrected.

- Line 25: suggested change: "In the model, nitrogen fixation was not able to adjust: : :": Added.

- Line 25. Here, it would help if the authors clarify why nitrogen fixation does not adjust to the high P levels. Because these are model results, this can be specified. We added a sentence to clarify this: "This is because low temperatures and iron limitation inhibited the uptake of the extra P and growth by nitrogen fixers in polar and lower latitude regions."(Line 27)

- Line 27: "this contrasts with" Corrected.

- Line 27. Here, the authors could clarify whether the palaeo reconstructions refer to model studies or reconstructions based on data or both and exactly how those results are different. We made here a very general statement, so we decided to remove this sentence.

- Line 49. Suggested change "the Earth has experienced" or "the Earth experienced" Corrected.

- Line 49. It's not clear what is meant by "climate OAE-like states". I would suggest to rephrase. We rephrased this sentence to be more specific: "Evidence in the palaeo record indicates that the Earth has experienced several OAEs with large-scale anoxia, euxinia and mass extinctions (Kidder and Worsley, 2010)."(Line 51)

- Line 66. What about increased inputs of Fe from continental shelves upon ocean deoxygenation? Could they alleviate the Fe limitation in the model? What is the uncertainty in the dust inputs? We just recognized that this sentence could imply that we simulate a dynamic dust cycle. Instead spatial dust limitation is based in UVic on present day observations. We decided to remove this sentence to not give this wrong

impression.

- Line 79. "and improved representations of" Corrected.

- Line 96: "as the switching point" Corrected.

- Line 98: "so that" (instead of "such that") Corrected.

- Line 111 and Table 1: are benthic P fluxes equated to burial? I don't see the term benthic flux in Table 1. The terms need to be used consistently. We refereed here to benthic P fluxes across the ocean-sediment boundary in general. To avoid confusion we replaced "benthic P fluxes" with "benthic P burial". In other text passages "benthic P fluxes" was replaced by "benthic P burial" or "benthic P release" respectively.

- Line 112: in a print, the purple and blue are very difficult to differentiate I am not sure, if you refer here to the color in Table 1 or to the general usage of purple and blue. However, we increased the color intensity of the purple in Table 1. We avoided using the combination of the colors green and red in all plots to improve the distinguishability of the colors for persons with green/red color blindness.

- Table 1: please add much more detail on the abbreviations in the text (or can you think of an easier notation?). Now it is very hard for the reader to keep the various model scenarios apart. Note that the sequence doesn't match the text and that the anthropogenic flux not quantified. In brackets short expressions were added to let the reader easier identify how the abbreviations are related to changes in the model formulations: Bur_res (i.e. restricted release or P reservoir); Bur_Dun (i.e. burial parameterization from Dunne et al. 2007); Weath0.05, Weath0.10, Weath0.15, Weath0.38 (i.e. the number represents the preindustrial weathering flux); Anthr (anthropogenic) For the model simulation Anthr we referred to Fig. 2, because we used a temporal variable weathering flux: "The weathering flux in simulation Anthr is variable over time (Fig. 2a)." (Table 1)

- Line 136-138. Please explain this section on the organic C burial better: it seems

contradictory that all organic C is remineralized but that there is still organic C burial. We adapted the text to clarify that we calculate a 'virtual' C burial to determine the P burial. In this model no C burial is applied: "Virtual is meant in the sense that there is no explicit burial of organic C in the model, and instead all organic C is remineralized in the deepest ocean layer. In this study we do not focus on changes in the C inventory and therefor BURC is just calculated to determine BENC in Eq. (2)." (Line 148)

- Line 258. "is essentially equal" Corrected.

- Lines 225-226. It is very well known already for a long time that total P in rivers can be mobilized in the coastal zone and forms a key input of P to the coastal zone. These are not new findings of Benitez-Nelson, Compton et al. and Ruttenberg as suggested here, so I would suggest to rephrase this sentence. We rephrased the sentence and added a reference to older studies: ". . .(see Fig. 1, Benitez-Nelson, 2000; Compton et al., 2000; Ruttenberg, 2003). These studies give a range of total P fluxes to the oceans, which are higher than interfered from dissolved inorganic P fluxes shown already in previous studies (e.g. Martin and Meybeck, 1979; Rao and Berner, 1993). . ." (Line 229)

- Same section: you could consider including a reference to the weathering & anthropogenic flux of P calculated from the Global News models (see the work of Seitzinger et al. 2010; Global Biogeochemical Cycles) and follow-up studies (e.g. Beusen et al. 2016; Biogeosciences) We cited in this paragraph Seitzinger et al. (2005) to reference to the Global News model.

- Lines 258. Change to "essentially" Corrected.

- Line 267. Change to "after the year" (note that "the" is missing before year in more places) Corrected.

- Line 327: Change to "a consequence" Corrected.

- Line 350. Change to "shown" Corrected.

- Line 358-359. This is an important piece of information that also should be given in

the abstract, see comment above. We added this information to the abstract. "This is because low temperatures and iron limitation inhibited the uptake of the extra P and growth by nitrogen fixers in polar and lower latitude regions."

- Don't start a sentence with "So that" Improved.

- Line 440-443. Needs rephrasing since benthic P release is known to be important in the coastal zone from both experimental field studies and modeling. Yes we agree with you. However, we focused here how a release of the actual inventory of sedimentary P could contribute to an increase in the oceanic P inventory. The deposition of fluvial particulate P to the sediment and its release was not simulated. We rephrased the text passage to state this more clear: "This could imply that benthic release of P is actually negligible in comparison to the weathering fluxes of P, but the UVic model does not resolve coastal processes such as the deposition of reactive particulate P from rivers on the continental shelves and its dissolution and release to the water column. For a more honest comparison of benthic and fluvial P fluxes, a more detailed representation of coastal processes would be necessary to simulate deposition and release of fluvial P from the sediments at the shelf. However, we can conclude that the actual local inventories of P are too small to sustain a positive benthic P feedback over several millennial."(Line 457)

- Line 459. The term "palaeo study" is vague. Please provide more information i.e. on the type of setting and time period. We provided more informations to the palaeo study in the text: "In the time period of the OAE1a and the OAE2, a substantial increase in N2-fixation was also inferred from measurements of sediment nitrogen isotope compositions typical for newly fixed nitrogen conditions and from high abundances of cyanobacteria indicated by a high 2-methylhopanoid index (Kuypers et al., 2004)."(Line 481)

- Line 460. "not able to compensate: : :." Unless you are underestimating the Fe input to the ocean. A few lines on the uncertainties there (e.g. shelf Fe input, dustfield) would be useful. We added a discussion in the result and discussion section: Section

4.2: "We acknowledge that in the current study we did not account for potential future changes in iron concentrations (from atmospheric deposition, shelf inputs) and that the lack of a fully prognostic iron model may lead to a different sensitivity of the response of diazotrophs." (Line 374) Discussion: "Sources of bioavailable Fe are still not well quantified and how these sources change under climate change is under debate (Hutchins et al., 2016; Mahowald et al., 2005). A more realistic representation of a dynamic iron cycle in UVic would affect N2-fixation in many areas of the global ocean (Fig. 8)."(Line 495)

- 'line 480. Could be changed to "on benthic P fluxes in this model is eventually: : :" These results really depend on how the N and Fe cycles are parameterized – should be discussed in a few more sentences in the main text. We added a short paragraph to section 5 and section 4.2 to discuss this (see also the last comment related to the Fe cycle): Section 4.2: "In our model, diazotrophs are limited by P and Fe and are not limited by N. Their growth rate, which depends on temperature being zero below 15°C, is slower relative to non-fixing phytoplankton. These characteristics allow them to succeed in warm, low-N and high–P environments that receive sufficient iron."(Line 362) "Similarly we did not account for the ability of phytoplankton to adapt to changing N:P ratios, that may affect marine biological productivity and in turn deoxygenation. These would require further studies."(Line 377)

Discussion: "As a next step it would be reasonable to investigate how different parameterizations of the N cycle and a full dynamic iron cycle will affect the utilization of the added P. For example benthic denitrification is not simulated in the UVic model. Model simulations showed for this century, that the enhanced denitrification in the water column could be compensated by less benthic denitrification (Landolfi et al., 2017), which could reduce the N-limitation and therefore enhance the effect of P fluxes on the biological pump."(Line 490)

- Fig. S2: I would suggest to move this figure to the main paper. We moved Fig. S2 to the manuscript and adapted the numbering.

[Figure]

---

## Author Comment (AC2) · 6 Feb 2019

First of all, we thank the reviewer very much for the constructive comments and advices, which helped to improve the manuscript a lot. Please find our reply below. Best regards, Tronje Kemena and co-authors

Reviewer comments: Kemena et al. present 12 long-term global warming simulations of the UVic Earth System Model of Intermediate Complexity to assess how projected P and O2 inventories depend on implemented weathering and sedimentary fluxes. The focus is on biogeochemical feedbacks, as the physical response is almost identical across the simulations. They suggest weathering fluxes contribute most to projected

increased P inventories. I consider this to be a welcome contribution to the field of long-term Earth system projections. I have several questions regarding the methodological approach and conclusions which should be considered to improve the readability and focus of the manuscript.

-Bur simulations: I find it hard to judge the "meaningfulness" of the Bur simulations given that they release P from an unlimited reservoir. The Bur_Res simulation seems to indicate an upper limit on the potential P release, which all other Bur simulations appear to surpass. So all other simulations release more P than can be assumed to be in the sediments, correct? Should the reservoir constraint not apply to all simulations? You might want to consider disqualifying those simulations a bit faster in the text and highlight the importance of this constraint a bit clearer, including the abstract and method/results section. In the Bur simulations, we investigate uncertainties in the release of P from the sediment in the future by analyzing different parameterizations for benthic P fluxes. These transferfunctions are used in various studies as a state of the art approach (e.g. Bohlen et al., 2012; Niemeyer et al., 2017; Wallmann, 2010), because transferfunctions are less cost intensive and easier to implement than full complex sediment models (Soetaert et al., 2000). In this study we like to push forward the development of these transferfunctions, but at the same time we like to point out how large uncertainties of such simple transferfunctions can be, therefore we believe that the results of all model simulations should be published.

Wallmann K (2010) Phosphorus imbalance in the global ocean? Global Biogeochem Cycles 24:1–12. doi: 10.1029/2009GB003643 Bohlen L, Dale AW, Wallmann K (2012) Simple transfer functions for calculating benthic fixed nitrogen losses and C:N:P regeneration ratios in global biogeochemical models. Global Biogeochem Cycles. doi: 10.1029/2011GB004198 Niemeyer D, Kemena TP, Meissner KJ, Oschlies A (2017) A model study of warming-induced phosphorus–oxygen feedbacks in open-ocean oxygen minimum zones on millennial timescales. Earth Syst Dyn 8:357–367. doi: 10.5194/esd-8-357-2017 Soetaert K, Middelburg JJ, Herman PMJ, Buis K (2000)

On the coupling of benthic and pelagic biogeochemical models. Earth-Science Rev 51:173–201. doi: 10.1016/S0012-8252(00)00004-0

-Weathering simulations: Why do these simulations not have a burial formulation? Of course, addition of P to an otherwise "closed" ocean just increase its P inventory. Would it not be important to assess the feedbacks associated with deposition and redissolution in this context, as increased P supply to the upper ocean stimulates NPP, export and deposition? I could not find a discussion on this matter in the manuscript.

In these simulations just anomalies ($W\_P=W\_(P,0)\cdot(f(NPP,SAT)-1)$ with $f(t=0)=1$) for weathering were applied. In a future study we could imagine to investigate above mentioned possible negative feedbacks. However, all simulations show an increase in NPP, export and the herewith associated increase in burial of P can be seen in all Bur simulations, especially in simulation Bur_high and Bur_Res with net global P loss at the end of the simulation. However in this study the low N availability is the predominant process that prevents the ocean from further deoxygenation (and not high burial rates in P). This can also be found in model simulations where such a negative feedback could by P are possible (Niemyer et al., 2017).

Niemeyer D, Kemena TP, Meissner KJ, Oschlies A (2017) A model study of warming-induced phosphorus–oxygen feedbacks in open-ocean oxygen minimum zones on millennial timescales. Earth Syst Dyn 8:357–367. doi: 10.5194/esd-8-357-2017

-It is not clear to me which simulation corresponds to the best estimate reported in the abstract and conclusion. Do you just add the result of the Weath0.15, Bur_Res and Anthr simulations? You may want to guide the reader a bit better here, and explain why these simulations are simply additive? That seems strange to me, as for instance, the Bur_Res simulation would suggest a W0=0.41 TmolPa-1, much higher than the 0.15 TmolPa-1 considered in Weath0.15.

We added these values together for following reasons: The anthropogenic input of P is prescribed and is extracted from Filippelli (2008). The weathering input depends

on environmental parameters and parameters of the weathering equations. The environmental parameters are just affected by the climate and therefore the changes in atmospheric CO2 concentrations. In our simulations, the climate develops in all simulations almost in the same way, so this would not affect the addition. However, you are right the export is much higher for simulations with larger oceanic P inventories. We hope that the additional removal of P will not affect the oceanic P inventory too much on this timescales. We shortly discussed your concerns in section 5: "In this simple addition of the P inventories we cannot account for feedbacks, which would appear in a fully coupled model. For such high P inventories we would expect larger suboxia and therefore more P release from sediments and at the same time a stronger export of organic P that lead to increased P burial."(Line 471)

Filippelli GM (2008) The Global Phosphorus Cycle: Past, Present, and Future. Elements 4:89–95. doi: 10.2113/GSELEMENTS.4.2.89

-Language: I find the language at times ambiguous. Most importantly, the usage of benthic fluxes, burial and sediment release appear confusing, and it's not clear whether gross or net exchange is meant. Please consider explicitly introducing these terms and using them consistently.

To avoid confusion we replaced "benthic P fluxes" in by "benthic P burial" or "benthic P release" if possible. Benthic P burial and benthic P release is defined in section 2: "P burial in the sediment (BURP) was determined in every grid box with sediment from the difference between the simulated detritus P rain rate to the sediment (RRP) and the benthic release of dissolved inorganic P from the sediment (BENP): ãĂŰBURãĂŮ_P=ãĂŰRRãĂŮ_P-ãĂŰBENãĂŮ_P (1) where RRP is the detritus flux from the ocean (in P units)." (Line 138) We used these definitions consistently throughout the manuscript. P burial/release leads to a net loss/gain of P in the global P inventory. We are not sure what do you meant with "gross exchange". "Benthic fluxes of P" is used as a more general term and it describes fluxes of P across the benthic boundary layer.

-Balance between presentation of P and O2 response: Even though deoxygenation is mentioned in the title, there is very limited mentioning of deoxygenation in the abstract and discussion. The most important driver of ocean deoxygenation appear to still be circulation changes, and the assessed biogeochemical feedbacks should be presented in this context. There are several papers worth citing/discussing in the introduction and discussion on the matter of long-term projections of ocean deoxygenation, e.g.: Battaglia & Joss 2018, ESD, Yamamoto et al. 2015, GBC, Schmittner et al. 2008, GBC As such, the modeled circulation response may be compared to other long-term projections.

The focus of this manuscript is to assess uncertainties in the projections of the P inventory and how this could affect deoxygenation. We agree that a reader of the title could expect a more general investigation of deoxygenation processes, therefore we decided to change the title of the manuscript to: "Ocean Phosphorus Inventory: Large Uncertainties in Future Projections on Millennial Timescales and its Consequences for Ocean Deoxygenation" Furthermore deoxygenation is always analyzed in relation to changes in the P inventory as it was already mentioned in the introduction: "Here, we build on this study and test the sensitivity of the marine P and O2 inventories in a climate change scenario on millennial timescales to different model formulations of P weathering and benthic fluxes."(Line 74) We removed the sentence from section 3 to avoid the impression that just uncertainties in P land-ocean and P sediment-ocean fluxes can be the drivers for the large range of P fluxes: "We found that the large range in P fluxes was not related to differences in the climate or atmospheric CO2 forcing, but rather to differences in parameterizations of P land-ocean (Sect. 3.1) and sediment-ocean (Sect. 3.2) interactions."

Specific comments:

Line 27: "this is in contrast to paleo reconstructions": not clear what is meant from the text We made here a very general statement, so we decided to remove this sentence.

Line 29: "more reliable projections of ocean deoxygenation": context of ocean deoxygenation does not emerge from the abstract. How do the biogeochemical feedbacks assessed compare to uncertainties in circulation changes? Introduction: centers round CO2-driven ocean deoxygenation. Please include recent literature on ocean deoxygenation, and the fact that circulation changes are crucial for ocean deoxygenation associated with global warming, not CO2 per se.

We completely agree with you, circulation changes are crucial for ocean deoxygenation too, but there are a series of factors e.g. changes in oxygen solubility, stratification, wind and upwelling, respiration, circulation and mixing processes that effect deoxygenation (Levin, 2019; Oschlies et al., 2018). We added a sentence in the beginning of the introduction: "Many different processes affect the oxygen balance in the ocean (e.g. oxygen solubility, stratification, respiration, circulation, Levin, 2019; Oschlies et al., 2018)." We focus here on very long time scales and here we were not able to find an answer in literature to the question: "How do the biogeochemical feedbacks assessed compare to uncertainties in circulation changes?". However, in this study we focus how uncertainties in benthic fluxes of P can affect the deoxygenation of the ocean and for this reason the temporal evolution of the ocean circulation is kept the same in all model simulations. In one paragraph of the introduction we focused on CO2, because the increase in atmospheric CO2 and consecutively climate warming is the most likeliest driver for OAEs. Actually changes in P supply (the same for circulation) are just a consequence of the CO2 induced global warming.

Levin LA (2018) Manifestation, Drivers, and Emergence of Open Ocean Deoxygenation. Ann Rev Mar Sci 10:229–260. doi: 10.1146/annurev-marine-121916-063359 Oschlies A, Brandt P, Stramma L, Schmidtko S (2018) Drivers and mechanisms of ocean deoxygenation. Nat Geosci 11:467–473. doi: 10.1038/s41561-018-0152-2

Line 57: "could": will? We replaced "could" with "will" as suggested.

2.3/2.4: Please consider presenting weathering first, consistent with presentation of

results/Table 1 We changed the order of the experiments in Table 1 with first the burial experiments and than the weathering experiments to be more consistent with the order in the "Model and Methods" section.

Line 126: "every grid box": every bottom grid box? We clarified the text here, because in the sub grid bathymetry benthic fluxes are not just limited to the bottom grid box: "every grid box with sediment"

Line 137: "all organic C is remineralized in the deepest ocean layer": statement must be wrong? Eq. 3a-c: potentially include z1000m on the respective lines for clarity. Eq. 4: Is there only a O2-feedback on P fluxes? Should the C burial/redissolution not also be O2-dependent? Potentially worthy of discussion.

We adapted the text to clarify that we calculate a 'virtual' C burial to determine the P burial. In this model no C burial is applied: "Virtual is meant in the sense that there is no explicit burial of organic C in the model, and instead all organic C is remineralized in the deepest ocean layer. In this study we do not focus on changes in the C inventory and therefor BURC is just calculated to determine BENC in Eq. (2)." (Line 148)

Line 170: Please add numbers in parenthesis.

We clarified the text passage. The burial is by definition one magnitude larger see also Eq. 3b, c. Factor 0.14 and 0.014. We slightly rephrased the sentence: "In the standard formulation, C burial is by definition one magnitude larger in slope and shelf regions compared to the deep ocean (see Eq. 3b, c)." (Line 168)

Line 183-200: Hard to understand. Examples below: We improved the paragraph as suggested (see below).

Line 187/188: "for the continental shelf and slope": how was this done for all other grid cells? We added a sentence to clraify this in the text: "In accordance to Flögel et al. (2011), release of P from the deeper ocean (>1000 m) cannot exceed the rain rate of organic P to the sediment. For the continental shelf and slope,..." (Line 191)

Line 194: "local inventory": what do you mean with this? Do all cells have this inventory? Or is this an upper limit for inventories globally?

Meant here is that 100% of the total solid P can be released. We rephrased the sentence and moved the introduction of RESP to the next sentence. As the local inventory is given in $\mu$mol cm-2 it scales also with increasing sediment coverage (for every grid cell with sediment coverage). 113 $\mu$mol cm-2 is the upper local inventory, we name it in the text now maximum local inventory (RESP,max):" Together, these assumptions convert to a maximum local inventory of total solid P in the active surface layer of RESP,max = 113 $\mu$mol cm-2 (Eq. 6a)." (Line 198)

Line 195/196: reservoir can be replenished, but excess P is permanently buried? Is the $\Delta$RESp the replenished P or excess P? Eq. 6: are the > and > correct?

Yes excess P is permanently buried. We replaced $\Delta$RESP by $\Delta$RESP/$\Delta$t to indicate that this is the change of RESP over time. The valid range of RESP, with 113 $\mu$mol cm-2 its upper limit, is defined in Eq. 6b.

Line 197: "depending on environmental conditions": what do you mean with this statement? Depending how? Earlier you mentioned local inventories of 113 umol/cm2?

RESP is the actual local P inventory and variable over time. During the spin-up simulation the local P inventories adapt to the environmental conditions like oxygen concentration or rain rate of P. I hope this is now clearer by introducing the maximum local inventory of P RESP,max and the actual local inventory of P RESP.

Line 277: "the way sediment P reservoirs are represented": if represented at all or not. 3.2: Preferably start section with PI RR (lines 295-299) and PI burial rates (lines 279- 283). Then, would strongly suggest changing the tone of this section, in that Bur simulations without a reservoir constraint are not realistic. Potentially exclude those runs from Figs.5-8, Figures are very busy anyway and lines are hard to tell apart. Or explain why those are considered for assessment of ocean deoxygenation, still.

This question is answered in the comment (-Bur simulations) of the reviewer.

Line 308/309: Please add more citations of long-term O2 projections.

We additionally cited Matear and Hirst (2003) as well as Shaffer et al. (2009).

Line 374: "recovered": strange language. O2 inventory is still increasing and simulation has not reached steady state yet.

We replaced "recovered" with "reached present day values again"

Line 377: How is AOU/O2sat calculated? Potentially discuss/mention Ito et al., 2004, GRL?

We added a description how AOU/O2sat is calculated and discussed Ito et al., 2004: "AOU is calculated from the difference between the O2 saturation concentration and the in situ oxygen concentration assuming that all ocean water leave the surface layer saturated in O2. The calculation of AOU is in general biased to higher values, because in polar regions surface water leaves the surface water in respect to oxygen in a undersaturated state due to reduced air-sea gas transfer inhibited by sea ice (Ito et al., 2004). In UVic this leads to an overestimation of AOU by 30% (Duteil et al., 2013). For a warming ocean sea ice cover reduces which converts into an underestimation of the AOU anomaly in Fig. 9c." (Line 387-393)

Line 390: Are preformed nutrients carried as explicit tracers? Please introduce how signal is separated.

We described the calculation of preformed nutrients in the figure caption of Fig. 10: "Preformed nutrients are calculated as the difference between remineralized and total nutrient content. The calculations assume that all ocean water leave the surface layer saturated in O2."

Line 397: "are likely": how come you are not sure about this?

We removed "likely".

Line 398: "global N inventory constant": does not appear to be the case in Fig. 9a?

We discuss here changes in the parameters for the time period from the beginning of the simulation to simulation year 2200. To be more clear we slightly rephrased the sentence: "Until the year 2200, changes in circulation and climate are likely the main cause for the reduction in preformed N and P in the Ref simulation since global N and P inventories were almost constant in this time period (Fig 9a, solid red and black line)." (Line 412)

Line 399: Physical response: potentially summarize physical response earlier, perhaps right after 4., as this is the baseline response of the scenario which applies to all variables?

Thank your for your suggestion. Here in this manuscript we mainly focus on changes in the P cycle and therefore we would prefer to discuss changes in the meridional overturning later.

Line 400: is this the max of the global meridional overturning?

Yes it is the maximum. We replaced "meridional overturning" by "meridional overturning maximum".

Line 402: "consistent with reduction of export": in section 4.1 you appear to conclude that warmer temperatures enhance remineralization in the shallower ocean which reduces export?

We do not agree with your thoughts. An increased remineralization rate could also lead to faster recycling of nutrients from the shallow ocean to the surface ocean. This could increase the export production, because of the faster "recycling" of nutrients from the shallow ocean to the surface in comparison of the recycling of nutrients from the deep ocean to the surface for lower remineralization rates. The stratification due to the continuous warming inhibits exchange of surface waters with the deep ocean and increases the residence time of water in the ocean. Until year 2200 highest rates of

warming appear in the ocean and lead to a strong stratification of the ocean, but after this $CO_2$ emission decline and the rise in surface temperatures of the ocean is much weaker.

Line 404-407: "speculate"? This reasoning, also in line 415, I do not understand. I would speculate that these changes are associated with older water masses. AOU and ideal age probably are highly correlated, indicating that more $O_2$ is consumed in older waters. See literature.

We completely agree with your thoughts and with your conclusions, which was also reflected by the text: "We speculate that a weaker overturning increased the residence time of water and nutrients in the surface ocean. Nutrients staying longer in the euphotic zone are with a higher probability biologically consumed. This implies more efficient utilization of nutrients and, hence, the reduction in preformed nutrients and an increase in AOU."(Line 420) I am sorry that I do not understand your concern could you please explain yourself a bit more.

Table 1: Bur_low and Bur_high: reference to equation (4) rather than (3)? Bur: potentially also add coefficients here for consistency with Bur_low, Bur_high

We improved the table as suggested.

Figure 2a: net flux? What are the step-like increases in the Bur simulations associated with? Those are also present in subsequent figures?

We mention this already in the manuscript: "In Bur, a rapid increase in the benthic P release appeared in areas where the water turned suboxic and thus drove a positive benthic feedback between P release, productivity and deoxygenation. A limited supply of P from the sediment (Bur_Res) dampens this feedback."(Line 298)

Figure S2/S3: might benefit from an improved aspect ratio.

We improved the aspect ratio.

---

## Author Response (AR2)

Dear reviewer,

Thank you again for your many helpful comments from both iterations. Please find our reply to your actual comments below.

Best regards,

Tronje Kemena and co-authors

Reviewers' comments:

Reviewer #1:
The authors improved their manuscript in response to the reviews at many places.

But I am under the impression that they chose not to incorporate or argue away many of my previous comments.
Please be more specific. We would need here more details to differentiate between comments, which were treated well enough and the ones, which need more attention so that we are able to improve the manuscript further.

For instance, it is not clear to me why the authors refuse to cite relevant recent literature on O2 projections (Battaglia & Joss 2018, ESD https://doi.org/10.5194/esd-9-797-2018, Yamamoto et al. 2015, GBC https://doi.org/10.1002/2015GB005181, Schmittner et al. 2008, GBC https://doi.org/10.1029/2007GB002953).
Thank you for your suggesting to add these literature. We are citing now Battaglia and Joss (2018) and Yamamoto et al. (2015) in the beginning of the introduction. Yamamoto et al. (2015) discuss the role of southern ocean ventilation for global oxygen levels, which was not mentioned in earlier versions of the manuscript. However, we would need model simulations with different CO2 emission scenarios to estimate such an effect in more detail and we would like to focus here on the effects of variations in the P inventory. We incorporated also a citation from Schmittner et al. (2008). In their study UVic simulated a tripling of the suboxic water volume following CO2 emissions of business as usual scenario supporting our results found in the control simulation.

In this sense, I guess I do not have much more to add.

Line 14: don't understand the addition "and hence without significant differences in climate and circulation". Emission driven scenarios may very well provoke different feedbacks and therefore produce different climatic responses. I don't see how this conclusion (hence…) emerges from the description of the scenarios.
We agree with you, that this is not a valid conclusion. We removed "hence" and we changed the wording of this sentence as in the following:
"In this study, we assessed the major uncertainties in projected P inventories and their imprint on ocean deoxygenation using an Earth system model of intermediate complexity for **the same** business-as-usual carbon dioxide ($CO_2$) emission scenario until the year 2300 and subsequent linear decline to zero emissions until the year 3000 and **without significant differences in climate and circulation among the model simulations**."
The anthropogenic radiative forcing (CO2) ranges from 4.7232 to 4.7652 W/m2 in simulation year 2245 (year of maximum atmospheric CO2 concentrations). The difference of 0.042 W/m2 is small in comparison to the total increase by 4.7 W/m2.

[Figure]

**Figure 1 Atmospheric CO2 concentrations for all model simulations**

[revised manuscript text omitted]

---

## Author Response (AR3)

Dear authors,
I return your manuscript again for minor revisions. After re-reading Reviewer #2 original review and your response, I can actually understand Reviewer#2's points that you did not respond well to the review.
For instance, the first comment of Reviewer #2 described the potential problem that some of the simulations release P from an unlimited reservoir. You responded by saying that you investigated uncertainties by using different parameterizations. This does not address the concern of the reviewer. When I read the second comment, I had a similar feeling that you could have addressed the point of the reviewer better, and could have accommodated this issue in your discussion section. This is just to give you two examples.
Also, the comment of including certain references was not meant to be simply included in a general statement at the beginning of the manuscript, but rather to contrast your results with studies that are close related. This gives me the impression that this manuscript needs more work to adequately deal with the reviewer comments. After all, they spend their time to think about your study and raised valid concerns, and my impression is that the current version of the manuscript does not adequately reflect these. When you resubmit your manuscript, please highlight the modifications in the revised text and provide a response.

Kind regards,
Axel Kleidon
Editor

**Dear Editor,**
**We realise that we could have addressed some issues raised by reviewer 2 better and more clearly. Here below we expand and clarify all of our unclear responses to reviewer 2 using bold font.**

Reviewer #2:
Kemena et al. present 12 long-term global warming simulations of the UVic Earth System Model of Intermediate Complexity to assess how projected P and $O_2$ inventories depend on implemented weathering and sedimentary fluxes. The focus is on biogeochemical feedbacks, as the physical response is almost identical across the simulations. They suggest weathering fluxes contribute most to projected increased P inventories. I consider this to be a welcome contribution to the field of long-term Earth system projections. I have several questions regarding the methodological approach and conclusions which should be considered to improve the readability and focus of the manuscript.

-Bur simulations: I find it hard to judge the "meaningfulness" of the Bur

simulations given that they release P from an unlimited reservoir. The Bur_Res simulation seems to indicate an upper limit on the potential P release, which all other Bur simulations appear to surpass. So all other simulations release more P than can be assumed to be in the sediments, correct? Should the reservoir constraint not apply to all simulations? You might want to consider disqualifying those simulations a bit faster in the text and highlight the importance of this constraint a bit clearer, including the abstract and method/results section.

Answers:

In the Bur simulations, we investigate uncertainties in the release of P from the sediment in the future by analyzing different parameterizations for benthic P fluxes. These transfer functions are used in various studies as a state of the art approach (e.g. Bohlen et al., 2012; Niemeyer et al., 2017; Wallmann, 2010), because transferfunctions are less cost intensive and easier to implement than full complex sediment models (Soetaert et al., 2000). In this study we like to push forward the development of these transferfunctions, but at the same time we like to point out how large uncertainties of such simple transferfunctions can be, therefore we believe that the results of all model simulations should be published.

-Wallmann K (2010) *Phosphorus imbalance in the global ocean? GBC 24:1–12. doi: 10.1029/2009GB003643*
-Bohlen L, Dale AW, Wallmann K (2012) *Simple transfer functions for calculating benthic fixed nitrogen losses and C:N:P regeneration ratios in global biogeochemical models. GBC. doi: 10.1029/2011GB004198*
-Niemeyer D, Kemena TP, Meissner KJ, Oschlies A. *A model study of warming-induced phosphorus–oxygen feedbacks in open-ocean oxygen minimum zones on millennial timescales. Earth Syst Dyn 8:357–367. doi: 10.5194/esd-8-357-2017*
-Soetaert K, Middelburg JJ, Herman PMJ, Buis K (2000) *On the coupling of benthic and pelagic biogeochemical models. Earth-Science Rev 51:173–201. doi: 10.1016/S0012-8252(00)00004-0*

**What we failed to clarify earlier is that an unlimited P release from sediments is the standard, current assumption of state-of-the-art model studies which don't have a fully prognostic sediment model (Bohlen et al., 2012; Niemeyer et al., 2017; Walmann, 2010). This is why we use this standard approach in our "base" simulations. However, with the new sensitivity simulations carried out in this study, we show that this standard approach may have very large uncertainties. We regard this as an important result and an improvement of this study. Thus we opted for keeping the simulations with the "standard" and not only the "new" (Bur_res) P release approaches to highlight this important improvement relative to earlier work and better highlighted this result in the manuscript (lines 345-346).**

-Weathering simulations: Why do these simulations not have a burial formulation? Of course, addition of P to an otherwise "closed" ocean just increase its P inventory. Would it not be important to assess the feedbacks associated with deposition and redissolution in this context, as increased P

supply to the upper ocean stimulates NPP, export and deposition? I could not find a discussion on this matter in the manuscript.

In these simulations just the anomalies ($W_P = W_{P,0} \cdot (f(NPP, SAT) - 1)$ $with$ $f(t = 0) = 1$) for weathering were applied. In a future study we could imagine to investigate above mentioned possible negative feedbacks. However, all simulations show an increase in NPP, export and the herewith associated increase in burial of P can be seen in all Bur simulations, especially in simulation Bur_high and Bur_Res with net global P loss at the end of the simulation. However in this study the low N availability is the predominant process that prevents the ocean from further deoxygenation (and not high burial rates in P). This can also be found in model simulations where such a negative feedback could by P are possible (Niemyer et al., 2017).

*Niemeyer D, Kemena TP, Meissner KJ, Oschlies A (2017) A model study of warming-induced phosphorus–oxygen feedbacks in open-ocean oxygen minimum zones on millennial timescales. Earth Syst Dyn 8:357–367. doi: 10.5194/esd-8-357-2017*

**Under preindustrial conditions the weathering flux is assumed to be in steady state and in equilibrium with the P burial rate. However, under transient simulations the weathering P flux is not compensated for by an equivalent P burial. The reviewer is right to say that this results in an increase in the P inventory with positive feedbacks on NPP and export production and on the expansion of OMZ volume as seen from Fig. 5 and Fig. 6.**
**The inclusion of burial in weathering simulations in an earlier study has been shown to be small relative to the increase in benthic release of P due to the feedback involving redox-sensitive benthic P fluxes associated with the expansion of OMZ (Niemeyer et al. 2017, Fig. S1). We have now discussed this in lines 399-402.**

-It is not clear to me which simulation corresponds to the best estimate reported in the abstract and conclusion. Do you just add the result of the Weath0.15, Bur_Res and Anthr simulations? You may want to guide the reader a bit better here, and explain why these simulations are simply additive? That seems strange to me, as for instance, the Bur_Res simulation would suggest a W0=0.41 TmolPa-1, much higher than the 0.15 TmolPa-1 considered in Weath0.15.

We added these values together for following reasons: The anthropogenic input of P is prescribed and is extracted from Filippelli (2008). The weathering input depends on environmental parameters and parameters of the weathering equations. The environmental parameters are just affected by the climate and therefore the changes in atmospheric $CO_2$ concentrations. In our simulations, the climate develops in all simulations almost in the same way, so this would not affect the addition. However, you are right the export is much higher for simulations with larger oceanic P inventories. We hope that the additional removal of P will not affect the oceanic P inventory too much on this timescales.

We shortly discussed your concerns in section 5: "In this simple addition of the P inventories we cannot account for feedbacks, which would appear in a fully coupled

model. For such high P inventories we would expect larger suboxia and therefore more P release from sediments and at the same time a stronger export of organic P that lead to increased P burial."
*Filippelli GM (2008) The Global Phosphorus Cycle: Past, Present, and Future. Elements 4:89–95. doi: 10.2113/GSELEMENTS.4.2.89*

**We agree that our "best estimate" concept is unclear. We now clarified that, assuming a linear combination of P inputs by weathering, anthropogenic activities and redox-sensitive benthic P release, we consider the combined effect of weathering parameters closest to present day, the model formulation with limited P reservoir and anthropogenic fluxes from Filippelli (2008). We have addressed this both in the abstract and conclusions (lines 18-21, 536-538).**

-Language: I find the language at times ambiguous. Most importantly, the usage of benthic fluxes, burial and sediment release appear confusing, and it's not clear whether gross or net exchange is meant. Please consider explicitly introducing these terms and using them consistently.

To avoid confusion we replaced "benthic P fluxes" in by "benthic P burial" or "benthic P release" if possible. Benthic P burial and benthic P release is defined in section 2:

"P burial in the sediment ($BUR_P$) was determined in every grid box with sediment from the difference between the simulated detritus P rain rate to the sediment ($RR_P$) and the benthic release of dissolved inorganic P from the sediment ($BEN_P$):

$$BUR_P = RR_P - BEN_P \qquad (1)$$

where $RR_P$ is the detritus flux from the ocean (in P units)."

We used these definitions consistently throughout the manuscript. P burial/release leads to a net loss/gain of P in the global P inventory. We are not sure what do you meant with "gross exchange".

"Benthic fluxes of P" is used as a more general term and it describes fluxes of P across the benthic boundary layer.

-Balance between presentation of P and O2 response: Even though deoxygenation is mentioned in the title, there is very limited mentioning of deoxygenation in the abstract and discussion. The most important driver of ocean deoxygenation appear to still be circulation changes, and the assessed biogeochemical feedbacks should be presented in this context. There are several papers worth citing/discussing in the introduction and discussion on the matter of long-term projections of ocean deoxygenation, e.g.: Battaglia & Joss 2018, ESD, Yamamoto et al. 2015, GBC, Schmittner et al. 2008, GBC As such, the modeled circulation response may be compared to other long-term projections.

The focus of this manuscript is to assess uncertainties in the projections of the P inventory and how this could affect deoxygenation. We agree that a reader of the title could expect a more general investigation of deoxygenation processes, therefore we decided to change the title of the manuscript to: "Ocean Phosphorus Inventory: Large Uncertainties in Future Projections on Millennial Timescales and its Consequences for Ocean Deoxygenation" Furthermore deoxygenation is always analyzed in relation

to changes in the P inventory as it was already mentioned in the introduction: "Here, we build on this study and test the sensitivity of the marine P and O2 inventories in a climate change scenario on millennial timescales to different model formulations of P weathering and benthic fluxes."(Line 74) We removed the sentence from section 3 to avoid the impression that just uncertainties in P land-ocean and P sediment-ocean fluxes can be the drivers for the large range of P fluxes: "We found that the large range in P fluxes was not related to differences in the climate or atmospheric CO2 forcing, but rather to differences in parameterizations of P land-ocean (Sect. 3.1) and sedimentocean (Sect. 3.2) interactions."

**We thank the reviewer for pointing to these studies investigating the role of ocean circulation on ocean deoxygentation. In this study we compare simulations with different biogeochemical settings but with virtually the same ocean circulation. Thus, the relative changes in ocean oxygen content among our model simulations can only be ascribed to ocean biogeochemistry and not circulation changes. Thus, we mention the important role of ocean circulation on oceanic O2 in the introduction (line 33-35), but highlight that our model experiments are designed to test the sensitivity of different biogeochemical P parameterizations (line 98-99; 514-515).**

Specific comments:

Line 27: "this is in contrast to paleo reconstructions": not clear what is meant from the text
We made here a very general statement, so we decided to remove this sentence.

Line 29: "more reliable projections of ocean deoxygenation": context of ocean deoxygenation does not emerge from the abstract. How do the biogeochemical feedbacks assessed compare to uncertainties in circulation changes? Introduction: centers round CO2-driven ocean deoxygenation. Please include recent literature on ocean deoxygenation, and the fact that circulation changes are crucial for ocean deoxygenation associated with global warming, not CO2 per se.
We completely agree with you, circulation changes are crucial for ocean deoxygenation too, but there are a series of factors e.g. changes in oxygen solubility, stratification, wind and upwelling, respiration, circulation and mixing processes that effect deoxygenation (Levin, 2019; Oschlies et al., 2018). We added a sentence in the beginning of the introduction: "Many different processes affect the oxygen balance in the ocean (e.g. oxygen solubility, stratification, respiration, circulation, Levin, 2019; Oschlies et al., 2018)." We focus here on very long time scales and here we were not able to find an answer in literature to the question: "How do the biogeochemical feedbacks assessed compare to uncertainties in circulation changes?".
However, in this study we focus how uncertainties in benthic fluxes of P can affect the deoxygenation of the ocean and for this reason the temporal evolution of the ocean circulation is kept the same in all model simulations. In one paragraph of the introduction we focused on CO2 , because the increase in atmospheric CO2 and consecutively climate warming is the most likeliest driver for OAEs. Actually changes in P supply (the same for circulation) are just a consequence of the CO2

induced global warming.

**As explained earlier, here we focus on how uncertainties in benthic fluxes of P can affect ocean deoxygenation by comparing simulations with different biogeochemical settings but with the same ocean circulation. Thus changes in ocean oxygen content results can only be ascribed to ocean biogeochemistry and not circulation changes (line 98-99; 514-515).**

Line 57: "could": will?
We replaced "could" with "will" as suggested.

2.3/2.4: Please consider presenting weathering first, consistent with presentation of results/Table 1
We changed the order of the experiments in Table 1 with first the burial experiments and than the weathering experiments to be more consistent with the order in the "Model and Methods" section.

Line 126: "every grid box": every bottom grid box?
We clarified the text here, because in the sub grid bathymetry benthic fluxes are not just limited to the bottom grid box: "every grid box with sediment"

Line 137: "all organic C is remineralized in the deepest ocean layer": statement must be wrong? Eq. 3a-c: potentially include z1000m on the respective lines for clarity. Eq. 4: Is there only a O2-feedback on P fluxes? Should the C burial/redissolution not also be O2-dependent? Potentially worthy of discussion.
We adapted the text to clarify that we calculate a 'virtual' C burial to determine the P burial. In this model no C burial is applied: "Virtual is meant in the sense that there is no explicit burial of organic C in the model, and instead all organic C is remineralized in the deepest ocean layer. In this study we do not focus on changes in the C inventory and therefor BURC  is just calculated to determine BENC  in Eq. (2)." (Line 148)

**In our simulations we assume that the C in the ocean-atmosphere is in steady-state, thus we don't account for organic C fluxes from land and explicit C burial (we do account for alkalinity fluxes however). We agree with the reviewer that accounting fro C burial may affect O2, due to reduced O2 consumption during remineralization. Based on our equations 3.1 and 3.2, we expect burial to have a small effect below 1000m, but a measurable effect above 1000m. We now improve the description (lines 175-177) and address this model caveat in line 589-590.**

Line 170: Please add numbers in parenthesis.
We clarified the text passage. The burial is by definition one magnitude larger see also Eq. 3b, c. Factor 0.14 and 0.014. We slightly rephrased the sentence:
"In the standard formulation, C burial is by definition one magnitude larger in slope and shelf regions compared to the deep ocean (see Eq. 3b, c)."

Line 183-200: Hard to understand. Examples below:
We improved the paragraph as suggested (see below).

Line 187/188: "for the continental shelf and slope": how was this done for all other grid cells?
We added a sentence to clarify this in the text:
"In accordance to Flögel et al. (2011), release of P from the deeper ocean (>1000 m) cannot exceed the rain rate of organic P to the sediment. For the continental shelf and slope,…"

Line 194: "local inventory": what do you mean with this? Do all cells have this inventory? Or is this an upper limit for inventories globally?
What we mean by "local inventory" is 100% of the total solid P can be released. To clarify this we rephrased the sentence and moved the introduction of $RES_P$ to the next sentence. As the local inventory is given in $\mu$mol cm$^{-2}$ it scales also with increasing sediment coverage (for every grid cell with sediment coverage). 113 $\mu$mol cm$^{-2}$ is the upper local inventory, we name it in the text now maximum local inventory ($RES_{P,max}$):" Together, these assumptions convert to a maximum local inventory of total solid P in the active surface layer of $RES_{P,max} = 113$ $\mu$mol cm$^{-2}$ (Eq. 6a)."

**We improved this description in lines: 225-236**

Line 195/196: reservoir can be replenished, but excess P is permanently buried? Is the ΔRESp the replenished P or excess P? Eq. 6: are the > and > correct?
Yes excess P is permanently buried. We replaced $\Delta RES_P$ by $\Delta RES_P/\Delta t$ to indicate that this is the change of $RES_P$ over time. The valid range of $RES_P$, with 113 $\mu$mol cm$^{-2}$ as an upper limit, is defined in Eq. 6b.

Line 197: "depending on environmental conditions": what do you mean with this statement? Depending how? Earlier you mentioned local inventories of 113 umol/cm2?
$RES_P$ is the "local P inventory" which is variable over time. This is set during the spin-up simulation, at equilibrium the local P inventories result from the environmental conditions like oxygen concentration or rain rate of P. To clarify this we introduced the maximum local inventory of P $RES_{P,max}$ and the actual local inventory of P $RES_P$.

Line 277: "the way sediment P reservoirs are represented": if represented at all or not. 3.2: Preferably start section with PI RR (lines 295-299) and PI burial rates (lines 279- 283). Then, would strongly suggest changing the tone of this section, in that Bur simulations without a reservoir constraint are not realistic. Potentially exclude those runs from Figs.5-8, Figures are very busy anyway and lines are hard to tell apart. Or explain why those are considered for assessment of ocean deoxygenation, still.
This question is answered in the comment (-Bur simulations) of the reviewer.

**An unlimited P release from sediments is the standard, current assumption of state-of-the-art model studies which don't have a fully prognostic sediment model (Bohlen et al., 2012; Niemeyer et al., 2017; Wallmann, 2010). This is why we use this standard approach in our "base" simulations. However, with the new sensitivity simulations carried out in this study, we show that this standard approach may have very large uncertainties. We regard this as an important result and an improvement of this study. Thus we opted for keeping the simulations with the "standard" and not only the "new" (Bur_res) P release approaches to highlight this important improvement relative to earlier work and better highlighted this result in the manuscript (lines 345-346).**

Line 308/309: Please add more citations of long-term O2 projections.

We additionally cited Matear and Hirst (2003) as well as Shaffer et al. (2009).

**As explained above, here we focus on how uncertainties in benthic fluxes of P can affect ocean deoxygenation by comparing simulations with different biogeochemical settings but with the same ocean circulation. Thus changes in ocean oxygen content results can only be ascribed to ocean biogeochemistry and not circulation changes (line 98-99; 514-515).**

Line 374: "recovered": strange language. O2 inventory is still increasing and simulation has not reached steady state yet.
We replaced "recovered" with "reached present day values again"

Line 377: How is AOU/O2sat calculated? Potentially discuss/mention Ito et al., 2004, GRL?
We added a description how AOU/O2sat is calculated and discussed Ito et al., 2004:
"AOU is calculated from the difference between the $O_2$ saturation concentration and the in situ oxygen concentration assuming that all ocean water leave the surface layer saturated in $O_2$. The calculation of AOU is in general biased to higher values, because in polar regions surface water leaves the surface water in respect to oxygen in a undersaturated state due to reduced air-sea gas transfer inhibited by sea ice (Ito et al., 2004). In UVic this leads to an overestimation of AOU by 30% (Duteil et al., 2013). For a warming ocean sea ice cover reduces which converts into an underestimation of the AOU anomaly in Fig. 9c."

Line 390: Are preformed nutrients carried as explicit tracers? Please introduce how signal is separated.
We described the calculation of preformed nutrients in the figure caption of Fig. 10:
"Preformed nutrients are calculated as the difference between remineralized and total nutrient content. The calculations assume that all ocean water leave the surface layer saturated in $O_2$."

**Preformed nutrients are not carried as explicit tracers in our simulations. They have been calculated using the standard technique described in Ito and Follows, 2005 (line 463-467)**

Line 397: "are likely": how come you are not sure about this?
We removed "likely".

Line 398: "global N inventory constant": does not appear to be the case in Fig. 9a?
We discuss here changes in the parameters for the time period from the beginning of the simulation to simulation year 2200. To be more clear we slightly rephrased the sentence:
"Until the year 2200, climate-driven circulation-slow down  contribute to the reduction in preformed N and P in the *Ref* simulation, and  N and P inventories are close to equilibrium during this time period (Fig 9a, solid red and black line)."

**We have clarified that significant N inventory changes occur only after 2200.**

Line 399: Physical response: potentially summarize physical response earlier, perhaps right after 4., as this is the baseline response of the scenario which applies to all variables?
Thank your for your suggestion. Here in this manuscript we mainly focus on changes in the P cycle and therefore we would prefer to discuss changes in the meridional overturning later.

**We have now discussed how the physical changes in ocean circulation and ventilation affects the distribution of O$_2$ and nutrients (line 474-478).**

Line 400: is this the max of the global meridional overturning?
Yes it is the maximum. We replaced "meridional overturning" by "meridional overturning maximum".

Line 402: "consistent with reduction of export": in section 4.1 you appear to conclude that warmer temperatures enhance remineralization in the shallower ocean which reduces export?
We do not agree with your thoughts. An increased remineralization rate could also lead to faster recycling of nutrients from the shallow ocean to the surface ocean. This could increase the export production, because of the faster "recycling" of nutrients from the shallow ocean to the surface in comparison of the recycling of nutrients from the deep ocean to the surface for lower remineralization rates. The stratification due to the continuous warming inhibits exchange of surface waters with the deep ocean and increases the residence time of water in the ocean. Until year 2200 highest rates of warming appear in the ocean and lead to a strong stratification of the ocean, but after this CO2 emission decline and the rise in surface temperatures of the ocean is much weaker.

**This apparent contradiction has been now now clarified in the text (393-395), despite the warming-driven enhanced remineralization, the**

**warming-driven intensification of ocean stratification leads to a decline in supply of nutrients to the surface layer. This is in line with earlier studies (eg: Bopp et al., 2013, Moore et al., 2013, Landolfi et al., 2017, Kvale et al., 2018)**

Line 404-407: "speculate"? This reasoning, also in line 415, I do not understand. I would speculate that these changes are associated with older water masses. AOU and ideal age probably are highly correlated, indicating that more O2 is consumed in older waters. See literature.
We completely agree with your line of thoughts and with your conclusions, which was also reflected by the text:
"We speculate that a weaker overturning increased the residence time of water and nutrients in the surface ocean. Nutrients staying longer in the euphotic zone are with a higher probability biologically consumed. This implies more efficient utilization of nutrients and, hence, the reduction in preformed nutrients and an increase in AOU."
I am sorry that I do not understand your concern could you please explain yourself a bit more.

**See lines 479-483 that reflect the reviewer comment.**

Table 1: Bur_low and Bur_high: reference to equation (4) rather than (3)? Bur: potentially also add coefficients here for consistency with Bur_low, Bur_high
We improved the table as suggested.

Figure 2a: net flux? What are the step-like increases in the Bur simulations associated with?
Those are also present in subsequent figures?
We discuss this in the manuscript:
"In *Bur*, a rapid increase in the benthic P release appeared in areas where the water turned suboxic and thus drove a positive benthic feedback between P release, productivity and deoxygenation. A limited supply of P from the sediment (*Bur_Res*) dampens this feedback."

**The step-like increase in P in the Bur simulations is is associated with changes in rapid changes in oxygen, leading to anoxic regions leading to extensive release of P from the sediments as seen from Fig, 2a. This P release has an effect on other parameters (NPP, Export and N2 fixation as can be seen from FIG. 6a, 6b, 7a). This is explained in line 332-334.**

Figure S2/S3: might benefit from an improved aspect ratio.
We improved the aspect ratio.

[revised manuscript text omitted]